# Nitrogen represses haustoria formation through abscisic acid in the parasitic plant *Phtheirospermum japonicum*

Anna Kokla [1], Martina Leso [1], Xiang Zhang[2], Jan Simura[3], Phanu T. Serivichyaswat[1], Songkui Cui [2], Karin Ljung [3], Satoko Yoshida[2] & Charles W. Melnyk [1✉]

Parasitic plants are globally prevalent pathogens that withdraw nutrients from their host plants using an organ known as the haustorium. The external environment including nutrient availability affects the extent of parasitism and to understand this phenomenon, we investigated the role of nutrients and found that nitrogen is sufficient to repress haustoria formation in the root parasite *Phtheirospermum japonicum*. Nitrogen increases levels of abscisic acid (ABA) in *P. japonicum* and prevents the activation of hundreds of genes including cell cycle and xylem development genes. Blocking ABA signaling overcomes nitrogen's inhibitory effects indicating that nitrogen represses haustoria formation by increasing ABA. The effect of nitrogen appears more widespread since nitrogen also inhibits haustoria in the obligate root parasite *Striga hermonthica*. Together, our data show that nitrogen acts as a haustoria repressing factor and suggests a mechanism whereby parasitic plants use nitrogen availability in the external environment to regulate the extent of parasitism.

[1] Department of Plant Biology, Linnean Center for Plant Biology, Swedish University of Agricultural Sciences, Almas allé 5, 756 51, Uppsala, Sweden. [2] Nara Institute of Science and Technology, Grad. School. Sci. Tech., Ikoma, Nara, Japan. [3] Umeå Plant Science Centre, Department of Forest Genetics and Plant Physiology, Swedish University of Agricultural Sciences, 90 183 Umeå, Sweden. ✉email: charles.melnyk@slu.se

Parasitic plants make up ~1% of all angiosperm species, some of which are devastating agricultural weeds that cause major agricultural losses each year[1–3]. Parasitic plants range from obligate parasites that completely depend on their host for survival to facultative parasites that survive without a host but parasitize when conditions are suitable[2,4]. Despite differences in their lifestyle, all parasitic plants form an invasive organ termed the haustorium[5] through which they penetrate the host and uptake water, nutrients, RNA, proteins, and hormones[5–10].

Many parasitic plants, particularly the obligate parasites, require perception of host-exuded compounds such as strigolactones to initiate germination. Perception of a second host-derived compound, known as haustorium inducing factors (HIFs), initiates haustorium formation in both obligate and facultative parasites. The first identified HIF was 2,6-dimethoxy-1,4-benzoquinone (DMBQ), originally isolated from root extracts of infected sorghum plants[11]. DMBQ can induce early stages of haustoria formation even in the absence of a host[11] in a wide range of parasitic plant species. In the facultative parasitic plant *Phtheirospermum japonicum*, perception of a nearby host via HIFs is followed by cell expansion and cell division at the haustorium initiation site, forming the characteristic swelling of the prehaustorium. Downstream signaling of HIFs requires reactive oxygen species (ROS) that accumulate in the haustorium after HIF perception[12]. Later, the developing haustorium attaches to the host and starts penetrating to reach the vascular cylinder of the host. Once the haustorium has reached the host's vasculature, it starts forming a xylem connection between itself and the host known as the xylem bridge[13–16]. These series of events lead to the establishment of the mature haustorium.

Despite recent advances in our understanding of haustorium development, we know little about how environmental conditions affect plant parasitism. Nutrient availability is an important factor affecting plant parasitism. Infestations of the agriculturally devastating obligate parasite *Striga* are often associated with poor soil fertility[17]. Low soil fertility is thought to impede host defences and exacerbate the damaging effects of infection[17]. In addition, low nutrient levels in the soil, particularly phosphate, promotes host secretion of strigolactones which enhances *Striga* germination and infection levels. Improving soil fertility can reduce the production of germination stimulants while also improving host defences and host tolerance[17–21]. However, nutrients might also have effects on the parasite beyond germination. For instance, the application of certain nitrogen compounds reduces *Striga* shoot development[22] whereas *P. japonicum* requires nutrient starvation to efficiently infect its hosts in vitro[8,13,14] and high phosphorous inhibits *Rhinanthus minor* growth[23]. Together, these data suggest that nutrients might play a role beyond improving host fitness or reducing parasite development.

Nutrient availability affects many aspects of plant development including germination, root growth, shoot growth and flowering[24–26]. High nitrate levels generally promote shoot growth and repress root growth, in part, through the action of plant hormones. In *Arabidopsis thaliana*, rice, maize and barley, nitrates increase cytokinin levels which move to the shoot meristems to promote cell divisions and growth[27–31]. Nitrates also inhibit auxin transport and modify auxin response to promote root initiation but inhibit root elongation[32]. The hormone abscisic acid (ABA) too plays a role; nitrate treatments increase ABA levels in *Arabidopsis* root tips[33] whereas ABA signaling is required for the inhibitory effects of high nitrates on root growth[34]. However, the mechanisms through which nutrient availability affects plant parasitism remain unknown.

Here, we show that nutrient-rich soils greatly reduce both root size and haustorial density in *P. japonicum*, and this effect is dependent specifically on nitrogen concentrations. Nitrogen application reduced ROS levels, blocked gene expression changes associated with haustoria formation and modified xylem patterning in the root. Nitrogen increased ABA levels and activated ABA responsive gene expression. Treating with ABA reduced haustoria initiation whereas inhibiting ABA biosynthesis or signaling reduced the inhibitory effect of nitrogen. Finally, we investigated the effects of nutrients in *Striga hermonthica* and found that similar to *P. japonicum*, nutrients decreased haustoria formation rates and infection rates, and this effect was specific to nitrogen and could be overcome by modifying phytohormone levels.

## Results

**Nitrogen inhibits haustoria development.** Previous work has demonstrated that nutrient-poor conditions are important for efficient *Striga* infestations and successful *P. japonicum* in vitro infections[8,13,35,36]. We tested whether successful *P. japonicum-Arabidopsis* infections in soil also required low nutrients by treating nutrient poor 50:50 soil:sand with or without fertilizer (51-10-43 N-P-K). *P. japonicum* shoot weights and heights were similar in both treatments, but root masses and haustorial density were higher under low nutrient conditions (Fig.1a–c; Supplementary Fig. 1a–c). To better understand the basis for reduced haustoria in high nutrient conditions, we grew 4–5-day old *P. japonicum* seedlings in vitro on water-agar or half-strength Murashige and Skoog medium (½MS)-agar (Supplementary Fig. 1d). Similar to fertilized soil, *P. japonicum-Arabidopsis* infections on ½MS-agar formed substantially fewer haustoria that also failed to form vascular connections with the host compared to those on water-agar (Fig.1d, e). To identify the compound(s) that caused haustoria arrest, we tested three of the major macroelements found in MS and tested one macroelement found in Gamborg's B5 medium at similar concentrations as those found in ½MS or Gamborg's medium. Agar media containing phosphate ($KH_2PO_4$ or $NaH_2PO_4$) or potassium ($KH_2PO_4$ or $KCl$) had little effect on haustoria formation, but agar media containing nitrogen including nitrates, ammonium or both ($KNO_3$, $NaNO_3$, $NH_4Cl$, $NH_4NO_3$) inhibited *P. japonicum-Arabidopsis* infections and xylem bridge formation similar to ½MS (Fig.1d, e, i). Infections on ½MS lacking nitrogen did not affect haustoria formation, xylem bridge formation or anatomy (Fig.1d, e, i; Supplementary Fig. 1d–g) indicating that nitrogen was sufficient and necessary to block haustoria formation. Nitrogen application led to a reduction of haustoria and xylem bridge formation in a 50 µM to 20.6 mM range of concentrations (Fig.1f, g; Supplementary Fig. 1e–g). However, plate xylem length, area and xylem bridge number were unaffected in haustoria that formed xylem bridges regardless of nitrogen treatment (Fig.1e, Supplementary Fig. 1e–g). To test whether nitrogen blocked infection by acting on the parasite or host, we applied $NH_4NO_3$ or ½MS to *P. japonicum* growing alone in the presence of the haustoria inducting factor DMBQ. Adding DMBQ to water or ½MS lacking nitrogen resulted in similar numbers of prehaustoria, whereas adding DMBQ to $NH_4NO_3$ or ½MS greatly reduced prehaustoria formation (Fig. 1h, i) indicating the effect of nitrogen on haustoria initiation did not depend on host infection.

**Haustoria formation induces widespread transcriptional changes.** To investigate how nitrogen availability affected haustoria formation in *P. japonicum*, we performed a time course RNAseq experiment of *P. japonicum* infecting *Arabidopsis* Col-0 in vitro on agar plates treated with water or 10.3 mM $NH_4NO_3$. We also included a treatment with 0.08 µM 6-benzylaminopurine (BA), a synthetic cytokinin, to test for similarities between $NH_4NO_3$ and BA transcriptional responses since previous studies

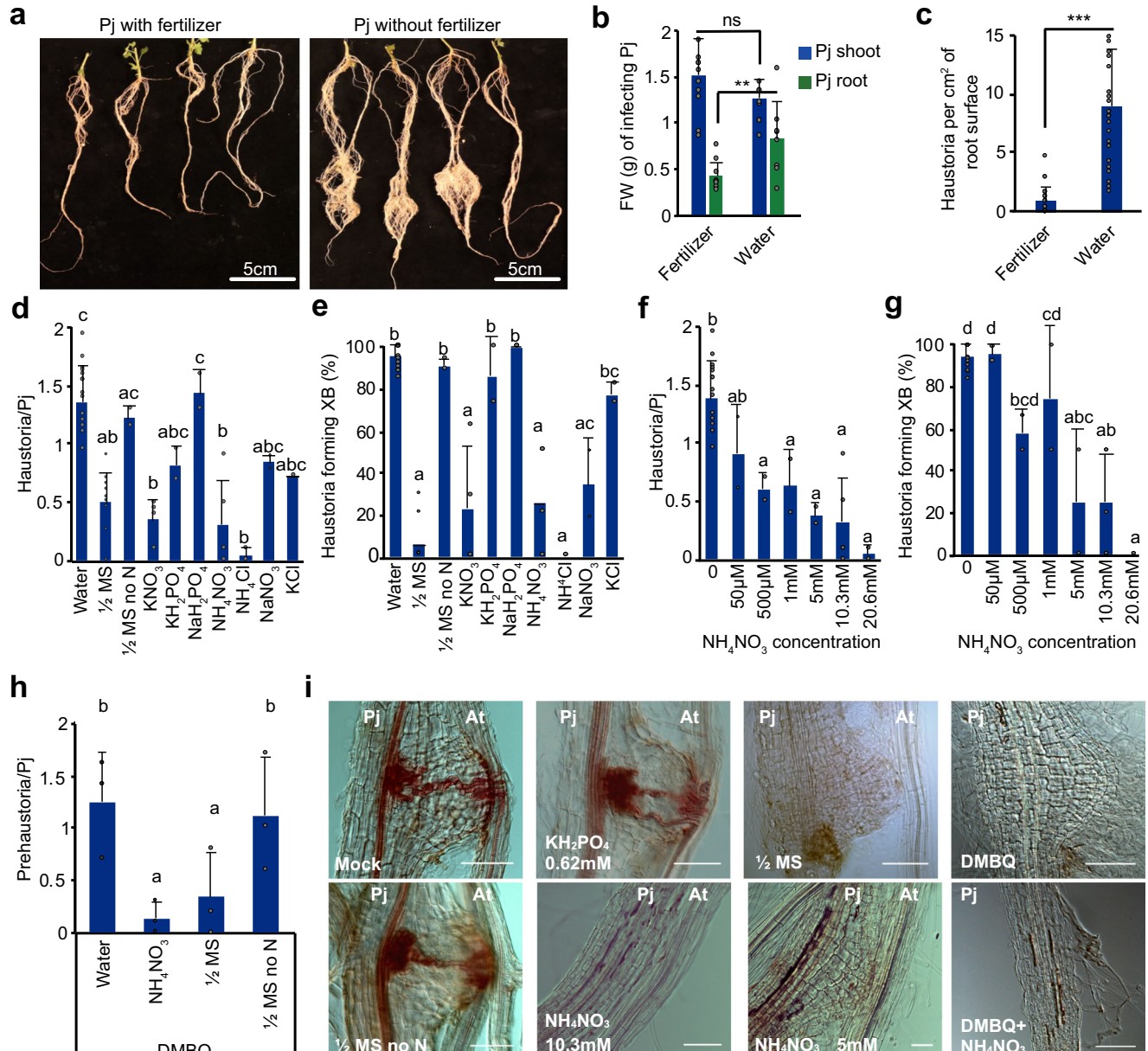

**Fig. 1 Nitrogen inhibits *P. japonicum* haustoria formation. a**, **b** Representative images and shoot and root fresh weight (FW) of *P. japonicum* during *Arabidopsis* infection with and without fertilizer (mean ± SD, n fertilizer= 11 plants, n water= 8 plants). **c** Haustoria numbers per cm$^2$ *P. japonicum* root during *Arabidopsis* infection with and without fertilizer (mean ± SD, n fertilizer= 24 images analyzed, n water= 37 images analyzed). **d**–**g** Haustoria numbers per *P. japonicum* seedling and xylem bridge formation percentage in in vitro *Arabidopsis* (Col-0) infections on ½MS (9 replicates), ½MS no N (2 replicates), 20.6 mM KNO$_3$ (4 replicates), 10.3 mM NH$_4$NO$_3$ (5 replicates), 0.62 mM KH$_2$PO$_4$ (2 replicates), 1.9 mM NaH$_2$PO$_4$ (2 replicates), 10.3 mM NH$_4$Cl (2 replicates), 10.3 mM NaNO$_3$ (2 replicates), 10.3 mM KCl (2 replicates) or various NH$_4$NO$_3$ (2 replicates) concentrations (mean ± SD, n = 20 plants per treatment per replicate). **h** Prehaustoria numbers per *P. japonicum* seedling with 10 μM DMBQ and half-strength MS, half-strength MS no N or 10.3 mM NH$_4$NO$_3$ (mean ± SD, n = 10 plants per treatment, 3 replicates). **i** Representative images of *P. japonicum* haustoria during *Arabidopsis* in vitro infections with various nutrient treatments (replicates as indicated in (**d**–**g**)). Scale bars 50 μm. **b**, **c** Asterisks represent \*$P < 0.05$, \*\*$P < 0.001$, \*\*\*$P < 0.0001$, Student's $t$ test, two tailed. Comparisons to the water treatment. **d**–**h** Different letters represent $P < 0.05$, one-way ANOVA followed by Tukey's HSD test. Source data provided.

have found that nitrogen treatments increase cytokinin levels in *Arabidopsis*, rice, maize and barley[27–30]. *P. japonicum* and *Arabidopsis* were physically aligned at time 0 to synchronize infections (Supplementary Fig. 1d) and tissues surrounding the root tips where haustoria normally emerge were collected at 0,12, 24, 48, 72 h post-infection (hpi) for the water treatment and 0,12, 24 hpi for the NH$_4$NO$_3$ and BA treatments (Fig. 2a). Additionally, as a control to distinguish transcriptional changes specific to haustorium formation, we included *P. japonicum* that grew without a

host on agar plates containing water, NH$_4$NO$_3$ or BA (Fig. 2a). As observed previously (Fig. 1), *P. japonicum* treated with NH$_4$NO$_3$ formed few to no prehaustoria whereas the water treatment resulted in successful haustoria formation. With the water treatment, we observed an increasing number of differentially expressed genes in infected samples compared to control samples as time progressed (Supplementary Fig. 2a). Co-expression analyses enabled us to classify genes into 8 clusters with distinct expression patterns during haustorium formation

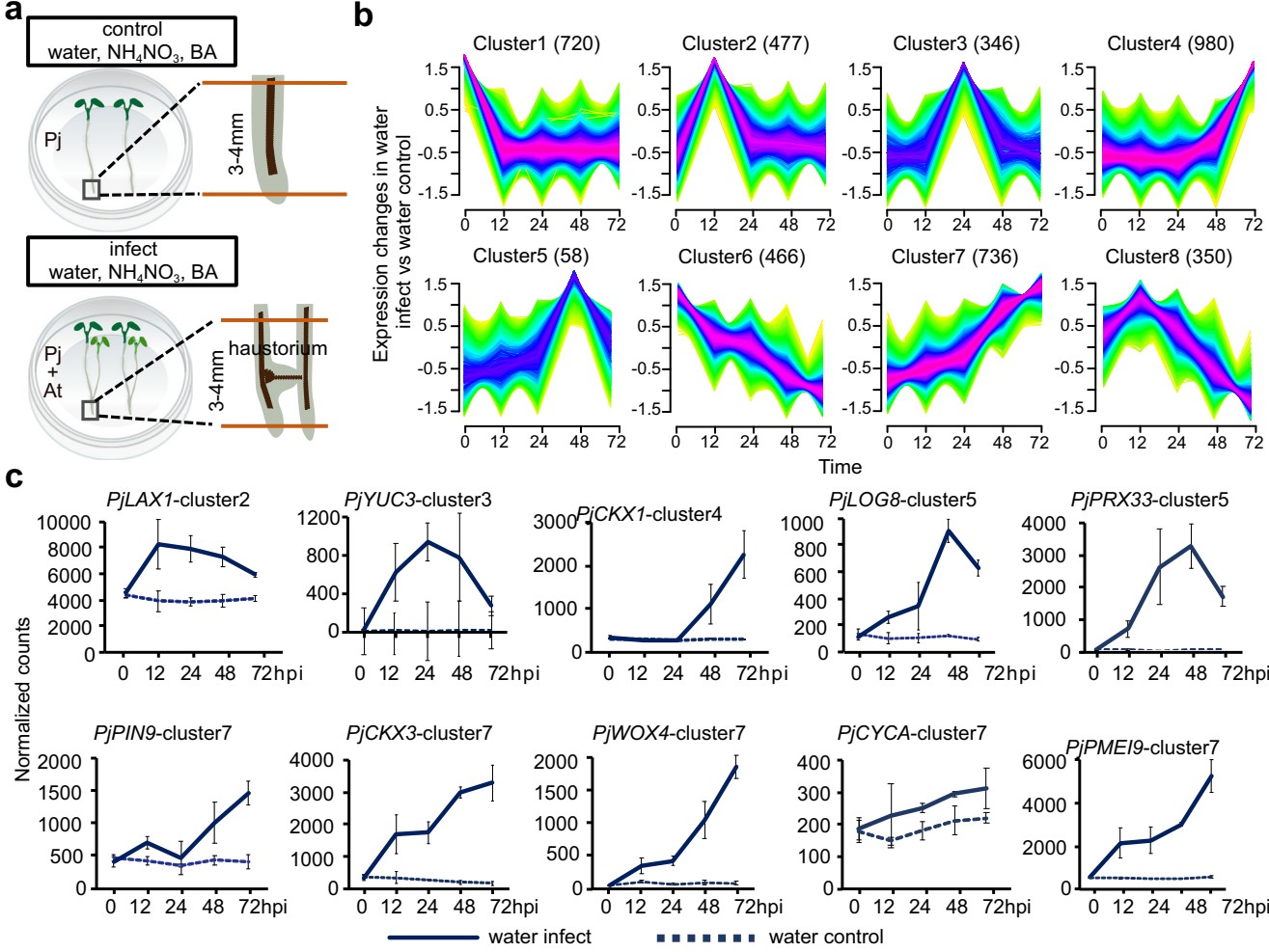

**Fig. 2 Transcriptomic changes during haustorium formation. a** Illustration describing the set-up for the RNAseq experiment. *P. japonicum* and *Arabidopsis* (infect) or *P. japonicum* alone (control) were placed on media containing water, 10.3 mM NH₄NO₃ or 0.08 μM BA; 3–4 mm of root tip or haustorium formation area was harvested at 0, 12, 24, 48, 72 hpi time points for the water treatment and 0, 12, 24 hpi time points for the NH₄NO₃ and BA treatments. **b** Clustering of differentially expressed genes in the water infect treatment based on their co-expression patterns over five time points; numbers in parentheses represent the number of genes in the cluster. **c** Normalized *P. japonicum* counts of *PjYUC3, PjLAX1, PjPIN9, PjWOX4, PjPMEI9, PjCKX1, PjCKX3, PjLOG8, PjCYCA, PjPRX33* over 5 time points in the water infect treatment (mean ± SD, *n* = 3 libraries).

(Fig. 2b, Supplementary Fig. 2b, Supplementary Data 1). Cluster 2, 3 and 8 whose gene expression peaked at early stages of haustoria formation (12 and 24hpi) had an over representation of genes that belong to Gene Ontology enrichment (GO) categories related to transcription, translation, signaling processes and cell expansion/ replication (Supplementary Fig. 3). Cluster 4, 5, and 7 whose gene expression peaked at later time points in haustorium formation (48 and 72hpi) had an over representation of genes that belong to GO categories related to response to oxidative stress, cytokinin metabolic process, fatty acid biosynthetic process, lignin, sucrose and carbohydrate metabolism (Supplementary Fig. 3). We looked at the expression of individual genes in our transcriptome and identified an upregulation of *P. japonicum* auxin-related *YUCCA3* (*PjYUC3*), *LIKE AUXIN RESISTANT 1* (*PjLAX1*), *PIN-FORMED 9* (*PjPIN9*), and cambium-related *WUSCHEL RELATED HOMEOBOX 4* (*PjWOX4*), genes whose expression has been previously observed to increase during *P. japonicum* infections[14,15,37] (Fig. 2c). Genes associated with cytokinin metabolism such as *P. japonicum CYTO-KININ OXIDASE 3* (*PjCKX3*), *CYTOKININ OXIDASE 1* (*PjCKX1*), *LONELY GUY 8* (*PjLOG8*), cell wall remodeling such as *PECTIN METHYLESTERASE INHIBITOR 9* (*PjPMEI9*), cell cycle such as *CYCLIN A* (*PjCYCA*) and ROS related such as *PEROXIDASE 33*

(*PjPRX33*) were upregulated as well (Fig. 2c) indicating substantial transcriptional reprogramming as the haustoria formed.

**Nitrogen inhibits genes associated with early haustorial development**. Nitrogen prevented haustoria formation (Fig. 1) so we looked at when this block occurs transcriptionally. We compared the transcriptional differences between infections on water and infections on NH₄NO₃ and found between 4000 and 6000 genes were expressed differently between treatments at each time point (Supplementary Fig. 2c–h) including *PjYUC3, PjWOX4,* and *PjPMEI9* whose expression was upregulated during successful haustoria formation on the water treatment but were not activated in the NH₄NO₃ treatment (Fig. 3a, b). Moreover, NH₄NO₃ treatment reduced the expression levels of cell cycle and ROS related genes (Fig. 4a, b). We tested this observation further and found EdU staining for cell division decreased (Fig. 4c) whereas dihydroethidium (DHE) and 2′,7′-dichlorodihydro-fluorescein diacetate (H₂DCFDA) staining for ROS accumulation reduced at the haustorium formation site at 24 hpi of NH₄NO₃ treatment (Fig. 4d–g) consistent with nitrogen acting early to block haustoria induction (Fig.1). We next compared NH₄NO₃

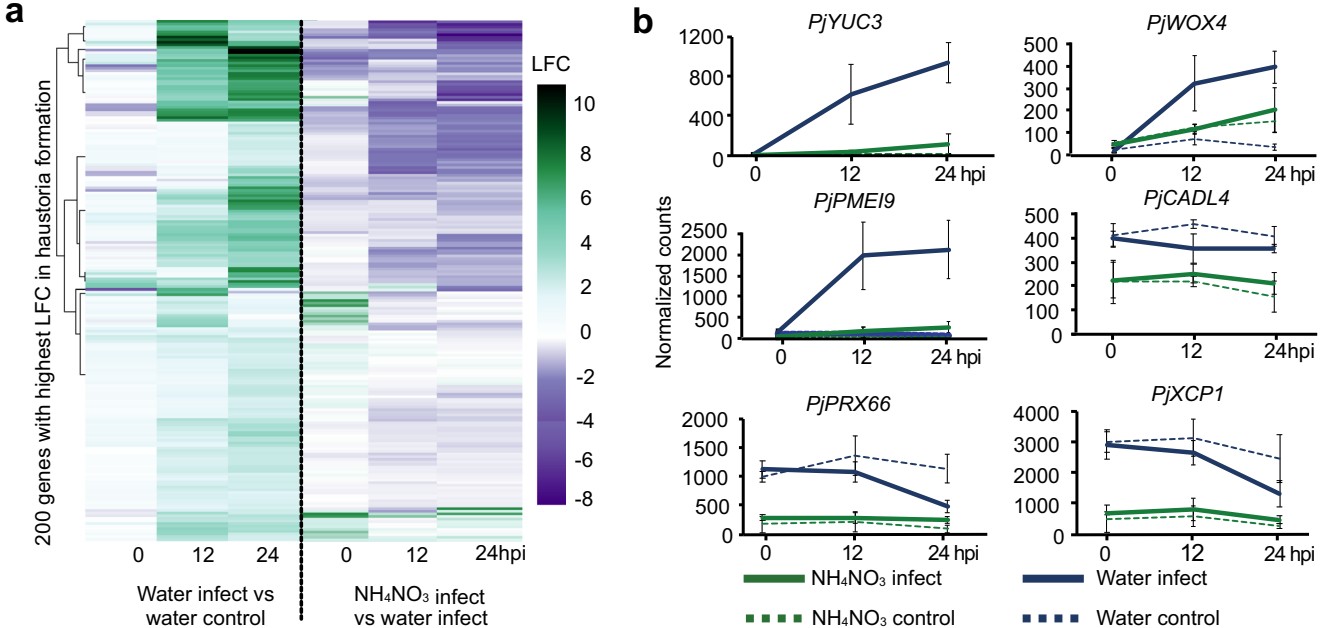

**Fig. 3 NH₄NO₃ treatment modifies *P. japonicum* gene expression. a** Heatmap of 300 *P. japonicum* genes with the highest log2 expression fold change during haustoria formation over three time points in the water infect vs water control and NH₄NO₃ infect vs water infect. **b** Normalized *P. japonicum* counts of *PjYUC3, PjWOX4, PjPMEI9, PjXCP1, PjCADL4, PjPRX66* over three time points shown for control and infect in the NH₄NO₃ and water treatments (mean ± SD, n = 3 libraries).

infect with NH₄NO₃ control and found fewer than 70 differentially expressed genes at any time point (Supplementary Fig. 2c) suggesting that very few infection-specific genes were upregulated in NH₄NO₃ infections. Consistent with this, less than 20 of these genes at each time point were also differentially expressed during water infections. Finally, we compared the NH₄NO₃ control to water control datasets to see which genes prior to infection might influence haustoria induction. We found a GO enrichment for cell wall and lignin-related genes downregulated in the NH₄NO₃ control compared to the water control (Supplementary Fig. 4a). Genes downregulated included xylem-related *P. japonicum XYLEM CYSTEINE PEPTIDASE 1* (*PjXCP1*), *LACCASE 11* (*PjLAC11*), *IRREGULAR XYLEM 3* (*PjIRX3*), *CELLULOSE SYNTHASE A4* (*PjCESA4*), *PEROXIDASE 66* (*PjPRX66*), quinone perception related genes *CANNOT RESPOND TO DMBQ LIKE 2* (*PjCADL2*) and *CANNOT RESPOND TO DMBQ LIKE 4* (*PjCADL4*)[38] and ROS related genes *PEROXIDASE 33* (*PjPRX33*) and *PEROXIDASE 25* (*PjPRX25*) (Fig. 3b, Supplementary Fig. 2i). Cytokinin-related GOs were also enriched in the genes downregulated by nitrogen (Supplementary Fig. 4a) and we found no substantial overlap between differentially expressed genes in the BA control and NH₄NO₃ control samples (Supplementary Fig. 5a). Together, these data suggested that nitrogen blocked the infection process at an early stage and nitrogen did not induce a substantial cytokinin response in *P. japonicum*.

**Nitrogen increases ABA levels in *P. japonicum*.** To further investigate how nitrogen arrests haustoria formation, we performed hormonal profiling on *P. japonicum* seedlings or mature roots infecting *Arabidopsis* with and without nitrogen treatment. In the parasite, levels of the active cytokinin trans-zeatin (tZ) and the cytokinin precursor trans-zeatin riboside (tZR) increased in successful infections, whereas in the host, levels of tZ and tZR increased in the presence of nitrates or successful infections, similar to previous studies[8,27–29,31] (Fig. 5a, Supplementary Fig. 6a, b). Neither jasmonic acid (JA), indole acetic acid (IAA),

gibberellic acid (GA) A1, ABA nor salicylic acid (SA) were substantially induced in the parasite by infection (Fig. 5a, b, Supplementary Fig. 6a, b). However, ABA and SA were significantly increased by NH₄NO₃ treatments in 50-day-old *P. japonicum* control and infect roots compared to water alone (Fig. 5b). In 20-day-old *P. japonicum* whole seedlings, ABA levels also increased in both infect and control NH₄NO₃ treated *P. japonicum* seedlings compared to water treatments (Fig. 5a). In the *Arabidopsis* host, ABA levels were also increased both by nitrogen treatments and by *P. japonicum* infection (Fig. 5a). This increase was dependent on host ABA biosynthesis since the increase was blocked in the ABA biosynthesis mutant *aba deficient 2-1* (*aba2-1*) (Fig. 5a). Cytokinin moves from *P. japonicum* to *Arabidopsis* during infections[8] but we found no evidence that ABA moved from parasite to host since *Arabidopsis* ABA levels in *aba2-1* infections were similar to not infected *aba2-1* plants (Fig. 5a). However, *P. japonicum* ABA levels were reduced in *aba2-1* infections compared to Col-0 infections, perhaps from reduced movement of host-derived ABA to the parasite or from the host reducing parasite ABA signaling or biosynthesis. We looked at our transcriptome analysis and found that *P. japonicum* genes homologous to *Arabidopsis* ABA responsive genes had increased expression levels in the NH₄NO₃ treatment compared to the water treatment for both infect and control tissues (Fig. 5c, Supplementary Fig. 4b, Supplementary Fig. 5e). This expression pattern was not seen for the same genes when comparing the BA control to water control, BA infect to water infect, or water infect to water control samples (Supplementary Fig. 4c, Supplementary Fig. 5b, f, g) suggesting the increased ABA response was specific to NH₄NO₃ treatment. Most cytokinin-related genes were not differentially expressed in the NH₄NO₃ infect compared to the water infect or NH₄NO₃ control compared to water control treatments—with some exceptions—further supporting our finding that NH₄NO₃ treatment in *P. japonicum* induces an ABA response rather than a cytokinin response (Fig. 5d, Supplementary Fig. 4d, e). However, cytokinin-related genes were differentially expressed during later time points in water infect compared to water control samples

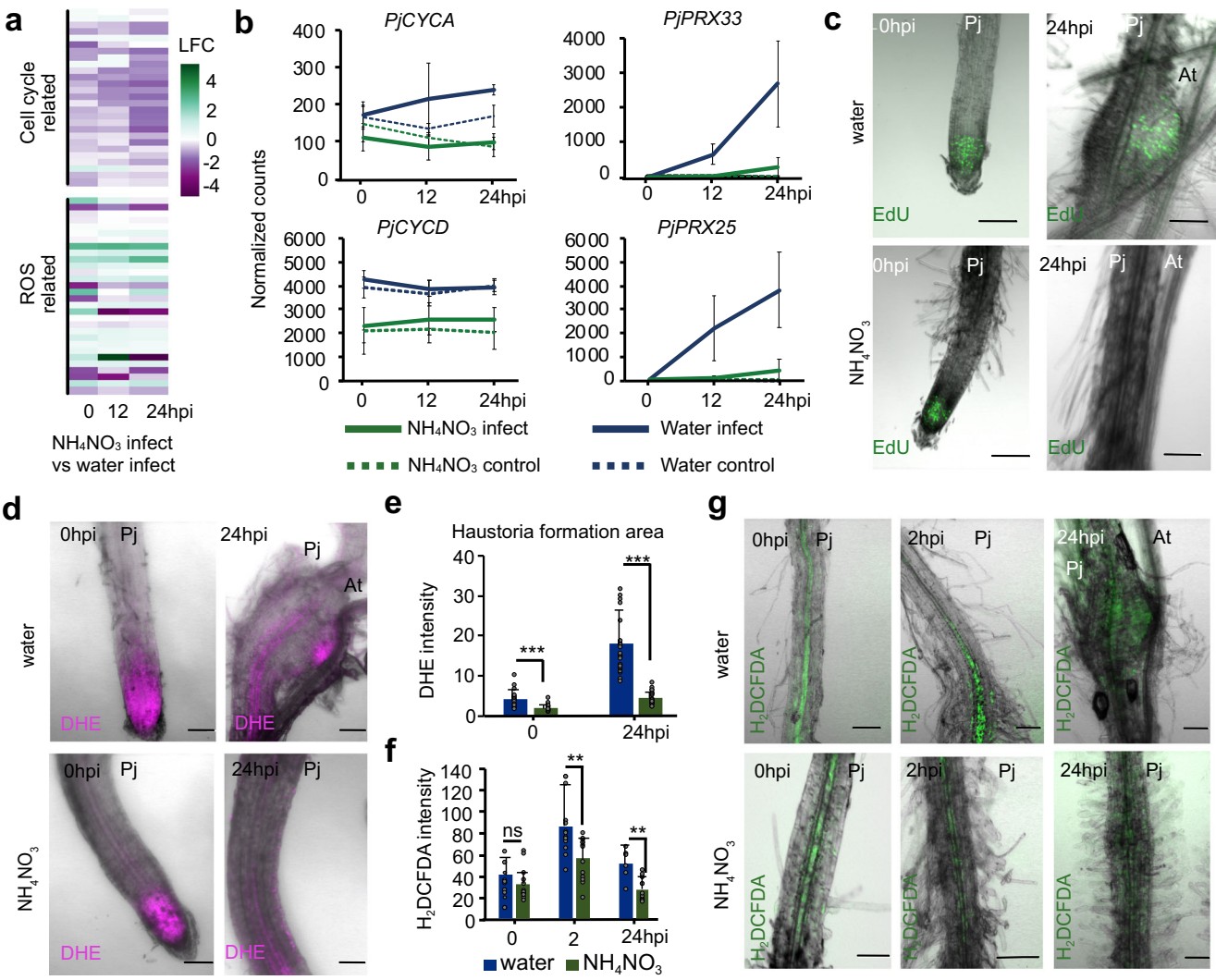

**Fig. 4 NH₄NO₃ reduces ROS accumulation and cell division in *P. japonicum*. a** Heatmap of 27 cell cycle and 30 ROS related *P. japonicum* genes in NH₄NO₃ infect compared to water infect RNAseq datasets. **b** Normalized *P. japonicum* counts of *PjCYCA*, *PjCYCD*, *PjPRX33*, *PjPRX25* over three time points shown for control and infect in the NH₄NO₃ and water treatment, bars represent mean ± SD. **c, d, g** Representative images of *P. japonicum* in vitro *Arabidopsis* Col-0 infections with water or 10.3 mM NH₄NO₃ at 0, 2 and 24 h post-infection (hpi) stained with EdU (c), dihydroethidium (DHE) (d) or 2′,7′-dichlorodihydrofluorescein diacetate (H₂DCFDA) (*n* = 20 plants per treatment per replicate, 2 replicates). Scale bars 100 μm. **e, f** DHE at 0 and 24 hpi (*n* = 22, water 0 hpi *n* = 17) and H₂DCFDA at 0, 2, 24 hpi (*n* = 12, 24 hpi water treatment *n* = 6) intensity in the *P. japonicum* haustorium formation area after water infect or NH₄NO₃ infect (mean ± SD; Comparisons to water treatments. Asterisks represent **P* < 0.05, ***P* < 0.001, ****P* < 0.0001, Student's *t* test, two tailed). Source data provided.

(Supplementary Fig. 5h), implicating cytokinin response in later haustorium development. To see whether the *P. japonicum* genes homologous to *Arabidopsis* ABA responsive genes were ABA responsive, we selected four upregulated genes in our NH₄NO₃ control RNAseq dataset and found by qPCR that transcript levels of three of them were significantly increased by exogenous ABA (Supplementary Fig. 5c). We tested the expression levels of these genes and one ABA biosynthesis homolog, *PjABA2*, in *P. japonicum* grown on various soil:sand ratios and found that the expression levels of *FRUCTOSE-BISPHOSPHATE ALDOLASE 2* (*PjFBA2*), *ABA INSENSITIVE 1* (*PjABI*) and *PjABA2* were lower in *P. japonicum* roots in nutrient poor soils (Supplementary Fig. 5d), suggesting that some ABA responses and ABA biosynthesis were downregulated in plants grown in nutrient poor soils, and conversely, upregulated in nutrient-rich soils (Supplementary Fig. 5d). SA levels also increased during nitrogen treatment in the parasite (Fig. 5a,b) but most *P. japonicum* genes homologous to *Arabidopsis* SA-responsive genes were not

differentially expressed in the NH₄NO₃ infect compared to the water infect treatment (Supplementary Fig. 5i). Together, these data suggest that nitrogen increased ABA levels and induced ABA responses in *P. japonicum*.

**ABA affects haustoria formation.** To investigate the role of ABA on haustorium formation in *P. japonicum*, we applied ABA exogenously using in vitro infection assays. ABA treated plants formed less haustoria than water treated plants (Fig. 6a, b). The application of fluridone, a chemical inhibitor of ABA biosynthesis, significantly reduced xylem bridge formation but did not affect haustoria formation (Fig. 6a–c). We reasoned that if nitrogen induced ABA to repress haustoria, we could overcome the inhibitory effects of nitrogen by blocking ABA biosynthesis with fluridone. Indeed, treating ½MS with fluridone increased haustoria numbers compared to ½MS alone but they remained intermediate to the water treatment (Fig. 6a–c). As a second

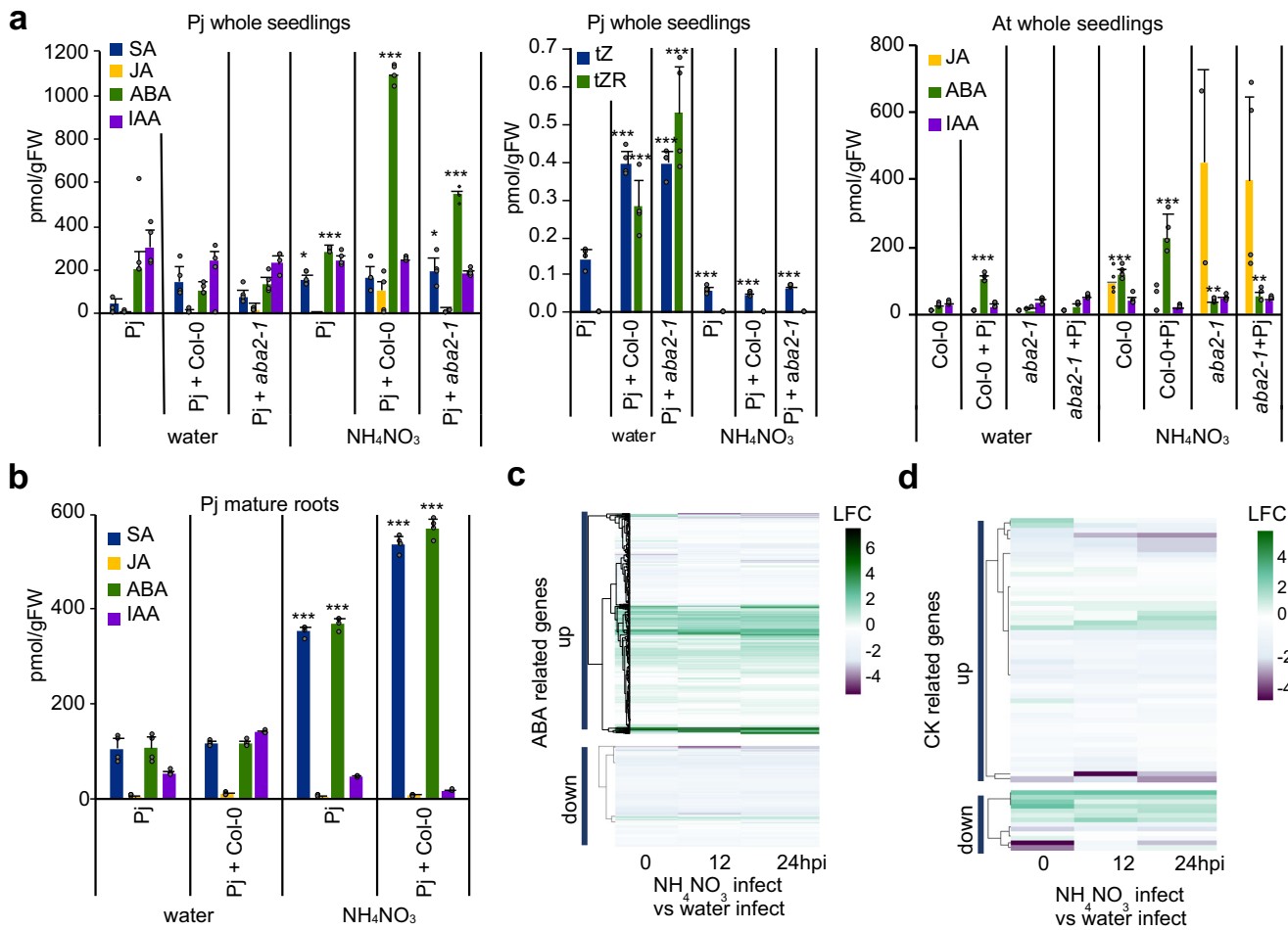

**Fig. 5 ABA levels increase during nitrogen treatment. a** Hormonal quantification of salicylic acid (SA), jasmonic acid (JA), abscisic acid (ABA), indole acetic acid (IAA), trans-zeatin (tZ) and trans-zeatin riboside (tZR) in *P. japonicum* whole seedlings and *Arabidopsis* (Col-0, *aba2-1*) whole seedlings treated with 10.3 mM NH₄NO₃ (mean ± SD, $n = 4$ plants per treatment, 4 replicates). **b** Hormonal quantification of SA, JA, ABA, and IAA in *P. japonicum* mature roots treated with 10.3 mM NH₄NO₃ (mean ± SD, $n = 4$ roots per treatment, 4 replicates). **c** Heatmap of the log2 fold change of 629 genes homologous to *Arabidopsis* ABA responsive genes (up or downregulated) for three time points in the NH₄NO₃ infected vs the water infected RNAseq dataset in *P. japonicum*. **d** Heatmap of the log2 fold change of 67 genes homologous to *Arabidopsis* cytokinin responsive genes (up or downregulated) for three time points in the NH₄NO₃ infect vs the water infect RNAseq dataset in *P. japonicum*. **a**, **b** Comparisons to respective control samples. Asterisks represent *$P < 0.05$, **$P < 0.001$, ***$P < 0.0001$, Student's *t* test, two tailed. Source data provided.

approach, we used the *Arabidopsis abi1-1* dominant mutant that represses ABA signaling in *Arabidopsis* and when expressed in *Nicotiana*[39,40]. We found that haustoria formation in non-transgenic hairy roots was suppressed by nitrogen but when we overexpressed *Atabi1-1* in *P. japonicum*, haustoria formation was unaffected by the presence of nitrogen (Fig. 6d–f). These results demonstrated that blocking ABA signaling in *P. japonicum* roots was sufficient to rescue the suppressive effects of nitrogen. We investigated host ABA pathways but *P. japonicum* infecting *Arabidopsis aba2-1* or *aba1-1C* did not have differences in haustoria and xylem bridge formation, suggesting that altering host ABA biosynthesis or signaling did not affect parasitism (Supplementary Fig. 6c, d). SA levels were also induced by NH₄NO₃ in *P. japonicum* and *Arabidopsis* (Fig. 5a, b, Supplementary Fig. 6a) so we tested the exogenous application of SA and found it decreased haustorial numbers but did not affect xylem bridge formation (Supplementary Fig. 6e–g, j, k). Thus, SA might act as a second signal to regulate haustoria, however, SA related genes were not differentially expressed by NH₄NO₃ treatment (Supplementary Fig. 5i) suggesting nitrogen does not induce an SA response.

Since ABA is important for various developmental processes including xylem formation[41], we analyzed the *P. japonicum* transcriptome in water infect compared to water control treatments and found that some *P. japonicum* genes homologous to *Arabidopsis* ABA responsive genes increased expression late during infection indicating they might have a relevant role during later stages of haustoria formation such as xylem bridge formation (Supplementary Fig. 5g). Exogenous ABA treatments did not increase xylem bridge formation, numbers or size (Fig. 6b, c, Supplementary Fig. 6e–i) but treatment with fluridone blocked xylem bridge formation (Fig. 6b, Supplementary Fig. 6i). Exogenous ABA application to *P. japonicum* was previously shown to enhance the number of differentiating xylem strands in primary root tips[41]. We repeated this assay but used nitrogen treatments on *P. japonicum* seedlings. ABA, NH₄NO₃ and ½MS all had a similar phenotype of increased xylem strand differentiation compared to water-only treatments (Supplementary Fig. 6l). These data showed that ABA treatment could phenocopy nitrogen treatment and that ABA played additional roles in both xylem bridge formation and also in modulating xylem patterning in response to nitrogen levels.

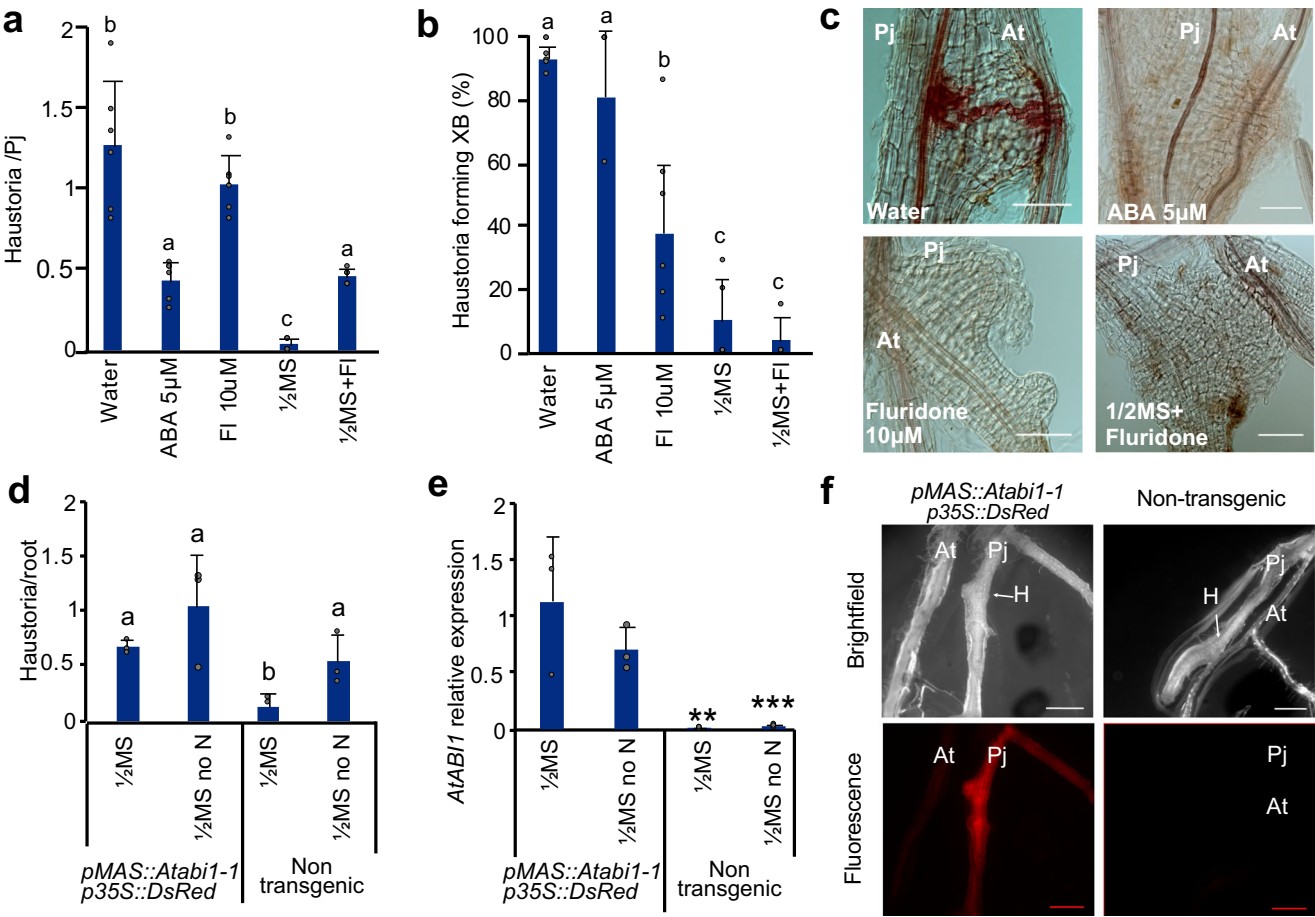

**Fig. 6 ABA represses *P. japonicum* haustoria formation. a, b** Haustoria number per *P. japonicum* seedling and xylem bridge formation percentage in in vitro *Arabidopsis* (Col-0) infections treated with ABA (6 replicates), fluridone (Fl) (6 replicates), ½MS (6 replicates), ½MS no N (6 replicates) or ½MS + fluridone (5 replicates) (mean ± SD, *n* = 20 plants per treatment per replicate). **c** Representative images of *P. japonicum* haustoria during *Arabidopsis* in vitro infection with ABA, fluridone, ½MS or ½MS + fluridone. Scale bars 50 µm. 5 replicates. **d** Haustoria number per *P. japonicum* hairy roots overexpressing *Atabi1-1* in in vitro infection assay with *Arabidopsis* Col-0 and ½MS or ½MS no N (mean ± SD, *pMAS::Atabi1-1 n* = 19 for ½MS and *n* = 24 for ½MS no N treatments, non-transgenic *n* = 30 for ½MS and *n* = 29 for ½MS no N treatments). **e** Expression levels of *Atabi1-1* in fluorescent and non-fluorescent hairy roots analyzed by qPCR (mean ± SD, *n* = 3 roots per treatment, 3 replicates). **f** Representative images of *P. japonicum* hairy roots in the ½MS no N treatment in brightfield and red fluorescence fields, the haustorium (H) is denoted by an arrow, scale bars 500 µm. 3 replicates. **a, b, d** Different letters represent *P* < 0.05, one-way ANOVA followed by Tukey's HSD test. **e** Asterisks represent **P* < 0.05, ***P* < 0.001, ****P* < 0.0001 compared to ½MS transgenic roots, Student's *t* test, two tailed. Source data provided.

**Nitrogen affects *Striga* infection rates.** Previous field studies have shown that *Striga* infestation is decreased after nitrate application[18,42]. We investigated the effect of nutrients upon *S. hermonthica* using in vitro infection assays with rice as the host. In the presence of $NH_4NO_3$, $KNO_3$, $KH_2PO_4$, or $NaH_2PO_4$, *Striga* infection rates were not significantly decreased 2 weeks after infection (Fig. 7a). However, 4 weeks after infection nitrogen application lead to a significant decrease in the percentage of *Striga* that infected its rice host and formed more than three leaves (Fig. 7a, d). *Striga* development was also hindered in the presence of nitrogen where the appearance of plants with 3–5 leaf pairs and more than 6 leaf pairs were decreased compared to the water treatment (Supplementary Fig. 7a). We tested whether this effect was due to improved host fitness or reduced *Striga* infectivity by treating *Striga* with DMBQ in the presence of nitrogen. Prehaustoria formation by DMBQ was significantly reduced in the presence of $NH_4NO_3$, $NH_4Cl$, or $KNO_3$ (Fig. 7b, c, Supplementary Fig. 7b) demonstrating that, like *P. japonicum*, early *Striga* haustoria formation is inhibited by high nitrogen. *Striga* is highly ABA[43] resistant, but nonetheless we tested exogenous application of ABA or fluridone and found they did not have an

effect on *Striga* haustoria formation in the presence or absence of nitrogen (Fig. 7e, Supplementary Fig. 7c, d). Auxin biosynthesis is important for haustoria formation[14] and we found that auxin-related genes were upregulated during *Striga*-rice infection indicating a possible role for auxin in promoting *Striga* haustoria formation (Fig. 7f). We applied exogenous auxin to nitrogen-grown *Striga* and found it could overcome the inhibitory effects of nitrogen (Fig. 7g), suggesting a role for auxin acting downstream of nitrogen. Exogenous auxin treatment did not affect haustoria formation in *P. japonicum* and did not rescue the haustoria inhibitory effect of ABA or ½MS (Supplementary Fig. 7e, f) consistent with *P. japonicum* and *Striga* using different hormone signaling pathways for nitrogen inhibition.

## Discussion

Here, we describe a mechanism whereby external nitrogen levels regulate haustoria formation in the facultative root parasite *P. japonicum* (Fig. 7h). This effect did not occur with phosphate or potassium and instead appeared highly specific to nitrogen in micromolar concentrations (Fig. 1f, g). A previous study showed

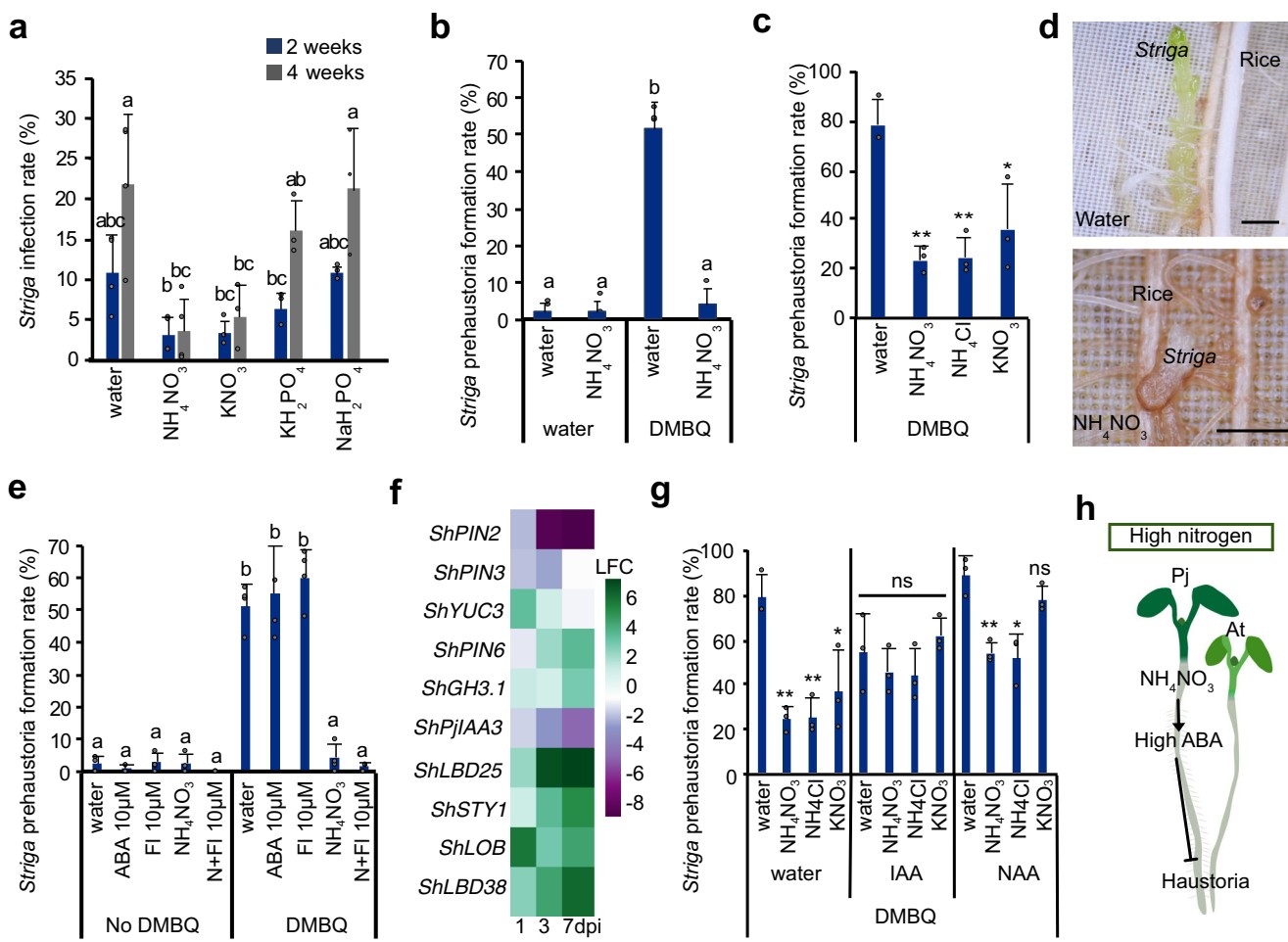

**Fig. 7 Nitrogen inhibits *Striga* infection rates. a** *Striga* infection rate of rice at 2 and 4 weeks with 20.6 mM KNO$_3$, 10.3 mM NH$_4$NO$_3$, 0.62 mM KH$_2$PO$_4$ or 1.9 mM NaH$_2$PO$_4$ (mean ± SD, $n = 8$ plants per treatment, 3 replicates). **b** Effect of 10.3 mM NH$_4$NO$_3$ on *Striga* prehaustorium formation induced by 10 μM DMBQ (mean ± SD, $n = 8$ per treatment, 3 replicates). **c** Effect of 10.3 mM NH$_4$NO$_3$, 10.3 mM NH$_4$Cl and 29.5 mM KNO$_3$ on *Striga* prehaustorium formation induced by 10 μM DMBQ (mean ± SD, $n = 8$ plants per treatment, 3 replicates). (d) Representative images of *Striga* infecting rice at 2 weeks after infection. Scale bars 1 mm. 3 replicates. **e, g** Effect of 10 μM ABA, 10 μM fluridone, 5 mM NH$_4$NO$_3$, 5 mM NH$_4$NO$_3$ + 10 μM fluridone, 500 nM NAA, 500 nM IAA, 10.3 mM NH$_4$NO$_3$, 10.3 mM NH$_4$Cl or 29.5 mM KNO$_3$ on *Striga* prehaustorium induction by 1 (**e**) or 10 (**g**) μM DMBQ (mean ± SD, $n = 8$ plants per treatment, 3 replicates). **f** Heatmap of *Striga* auxin-related gene expression during rice infection. **h** Graphical representation of a putative model of nitrate-ABA mediated haustoria regulation. **a, b, e** Different letters represent $P < 0.05$, one-way ANOVA followed by Tukey's HSD test. **c, g** Asterisks represent *$P < 0.05$, **$P < 0.001$, ***$P < 0.0001$ compared to water treatments, Student's $t$ test, two tailed. Source data provided.

that increased nitrogen supply to *Medicago sativa* also reduced *P. japonicum* parasitism[44], consistent with our results. We propose that local nitrogen supply at the site of infection has a suppressive effect upon the parasite in addition to nitrogen's proposed roles to stimulate host defenses and reduce host HIF production. *Striga* haustoria formation and infection rates were also inhibited by nitrogen (Fig. 7a–c) which is consistent with previous observations that external nitrogen reduces *Striga* growth[22,42]. However, our results point to an earlier additional role for nitrogen by preventing haustoria to develop beyond an initial swell (Fig.1h, Supplementary Fig. 7b). DHE and H$_2$DCFDA staining revealed nitrogen reduced ROS levels in *P. japonicum* roots prior to infection (Fig. 4), whereas quinone perception related genes *PjCADL2* and *PjCADL4* were downregulated by nitrogen which all could negatively affect HIF perception or downstream HIF signaling[12]. Alternatively, starvation could increase ROS and *PjCADL* levels to induce competency for HIF perception and haustoria elongation. As such, the observed reduction in *Striga* infestations in nutrient-rich fields[18] could be from a combination of reduced host germination stimulant

production and our findings that nitrogen reduced haustoria formation and *Striga* growth. Our results also suggest a conserved role for nitrogen acting as a haustoria repressing factor in both facultative and obligate Orobanchaceae family members.

Beyond parasitism, nitrogen has strong effects upon plant root architecture and organogenesis. In *Arabidopsis*, mild nitrogen deficiency enhances lateral root elongation, whereas uniform high nitrogen levels repress lateral root development[45–47]. In nodulating plants, high nitrogen levels in the environment repress nodule formation in *Medicago truncatula*, soybean and alfalfa through a regulatory mechanism involving multiple hormones and peptides[48–52]. Our data and these previous findings suggest a common regulatory theme whereby low nitrogen levels promote organ growth to uptake additional nutrients, whereas high nitrogen levels repress organ growth to avoid unnecessary resources spent on nutrient acquisition.

In species like *Arabidopsis*, maize, rice and barley, high nitrates increase cytokinin levels and these are important for root development and shoot growth[27–30]. We found host and parasite cytokinin levels were strongly induced by infection (Fig. 5a,

Supplementary Fig. 6a), consistent with previous findings[8], but nitrogen itself did not increase tZ levels, tZR levels or induced a strong cytokinin response in young *P. japonicum* and had a mixed effect in mature *P. japonicum* where tZ levels increased but tZR levels decreased upon nitrate treatment (Supplementary Fig. 6b). Notably, a recent study found that nitrogen treatment of *Lotus japonicus* inhibited cytokinin biosynthesis, reduced cytokinin levels and reduced nodule formation[53]. We saw a similar situation in young *P. japonicum* since nitrogen treatment reduced trans-zeatin levels and did not induce a strong cytokinin response (Fig. 5a, d). Thus, both *P. japonicum* and *Lotus* appear to increase cytokinin levels in response to successful haustoria or nodule formation[8,54] yet do not necessarily increase cytokinin in response to high nitrogen. This situation differs from many other flowering plants and might be a convergent strategy to use cytokinin to signal successful symbiosis rather than nutrient abundance.

ABA plays an important role in parasitic plants and we observed that nitrogen increased ABA levels in *P. japonicum* independently of infection, while *P. japonicum* infection increased ABA levels in *Arabidopsis* (Fig. 5a). ABA levels are known to increase in both *Rhinanthus minor* and *Cuscuta japonica*, as well as their hosts, after infection[55,56]. *Striga* parasitism also increases host ABA levels in tomato and maize, and commonly induces symptoms in the host mimicking drought stress[57,58]. The increase in ABA we observed in the host *Arabidopsis* was likely due to a stress or defense response rather than movement from the parasite, however, increases in ABA levels in the parasite may have come in part from the host. Such ABA increases in the parasite appeared biologically relevant since treatments with exogenous ABA reduced haustoria numbers whereas perturbing ABA biosynthesis or ABA signaling in *P. japonicum* chemically or genetically overcame nitrogen inhibition (Fig. 6). These results suggest that nitrogen regulated haustoria formation in part via increasing ABA levels and ABA response which in turn repressed early stages of haustoria development including cell division. *Striga* and *Cuscuta* are highly insensitive to ABA[43,59] and *Striga* did not respond to ABA in our assays (Fig. 7, Supplementary Fig. 7), indicating that these species likely use additional mechanisms for nitrogen-induced haustoria repression such as modifying auxin response which differed from the situation in *P. japonicum* (Supplementary Fig. 7e, f). Other factors including SA or proteins known to affect lateral root or nodule formation likely also play a role in haustoria regulation.

Our assays revealed several developmental roles for ABA in *P. japonicum*. ABA was important for haustoria inhibition and ABA treatment produced haustoria that were underdeveloped or did not attach well, likely explaining the partial reduction in xylem bridge formation from ABA treatment (Fig. 6). ABA was also important for xylem development since chemical inhibition of ABA biosynthesis led to reduced xylem bridge formation whereas nitrogen and ABA treatments induced early xylem differentiation in the primary root tip (Fig. 6b, Supplementary Fig. 6l). However, nitrogen also reduced the expression of xylem-related genes in the haustoria and surrounding tissues (Fig. 3, Supplementary Fig. 2) which might relate to differing roles for ABA both promoting xylem differentiation but also inhibiting haustoria formation.

In nodulating plants, such as *Lotus japonicus*, *Trifolium repens* and *Medicago truncatula*, ABA acts as a negative regulator of nodules by repressing nod factor signaling and cytokinin responses[60–62]. Exogenous application of ABA blocks the early stages of infection in *Lotus japonicus*[61], similar to the situation we observe with haustoria in *P. japonicum*. We propose that at least some parasitic plants and legumes share another common regulatory theme whereby ABA inhibits symbiotic organ formation. However, more work will be required to investigate these parallels including whether nitrogen induces ABA in legumes and whether ABA inhibits HIF signaling in parasitic plants. Given that legumes and most parasitic plants are distantly related, it begs the question of whether such ABA and cytokinin regulatory features might be an important adaptation for symbiotic nutrient acquisition.

## Methods

**Plant materials and growth conditions.** *P. japonicum* (Thunb.) Kanitz ecotype Okayama seeds harvested in Okayama and Karuizawa, Japan were used for our experiments[63]. *Arabidopsis* ecotype Columbia (Col-0) accession was used as *Arabidopsis* wild-type (WT). *Arabidopsis aba2-1* and *abi1-1C* were published previously[64,65]. For in vitro germination, seeds were surface sterilized with 70% (v/v) EtOH for 20 min followed by 95%(v/v) EtOH for 5 min then left to air-dry to remove remaining EtOH. The seeds were then sown on petri dishes containing ½MS medium (0.8% (w/v) plant agar, 1% (w/v) sucrose, pH 5.8). After overnight stratification in the dark and 4 °C, the plants were transferred to 25 °C long-day conditions (16-h light:8-h dark and light levels 100 $\mu$mol m$^{-2}$ s$^{-1}$).

*Striga hermonthica* (Del.) Benth seeds were kind gifts provided by Dr A. G. T. Babiker (Environment and Natural Resources and Desertification Research Institute, Sudan). Rice seeds (*Oryza sativa* L. subspecies *japonica*, cvs Koshihikari) used in this study were originally obtained from National Institute of Biological Sciences (Tsukuba, Japan) and propagated in the Yoshida laboratory. The *Striga hermonthica* seeds were sterilized with a 20% (v/v) commercial bleach solution for 5 min and washed thoroughly with sterilized water on a clean bench. After that, these surface-sterilized *Striga* seed were placed in 9 cm petri dishes with moisturized glass fiber filter paper (Whatman GF/A) and conditioned at 25 °C in the dark for 7 days. The conditioned *Striga* seeds were treated with 10 nM Strigol[66] for 2 hours prior to rice-infection treatments. For haustorium induction assays, the conditioned *Striga* seeds were treated with 10 nM Strigol at 25 °C for 1 day in the dark before starting incubation in various nutrient media with or without DMBQ and hormones for 24 h in dark condition.

Rice seeds were de-husked and sterilized with a 20% (v/v) commercial bleach solution (Kao Ltd., Japan) for 30 min with gentle agitation. The rice seeds were then washed thoroughly with distilled water and placed on filter papers in 9 cm petri dishes filled with 15 mL sterilized water in a 16-h light/8-h dark cycle at 26 °C for 1 week.

**In vitro infection assays with *P. japonicum*.** Four to five days old *P. japonicum* seedlings were transferred for three days to nutrient-free 0.8% (w/v) agar medium or 0.8% (w/v) agar medium supplemented by nutrient or hormone treatment: ½MS, ½MS no N, 20.6 mM KNO$_3$, 50 $\mu$M-20.6 mM NH$_4$NO$_3$, 0.62 mM KH$_2$PO$_4$, 1.9 mM NaH$_2$PO$_4$, 10.3 mM NH$_4$Cl, 20.6 mM NaNO$_3$, 5 $\mu$M ABA, 10 $\mu$M Fluridone, 5 $\mu$M SA, 500 nM NAA, or 10.3 mM KCl. Five days old *Arabidopsis* seedling were aligned next to and roots place in contact with these pre-treated *P. japonicum* roots for infection assays. Haustorium formation and xylem bridge development were measured at seven days post infection using a Zeiss Axioscope A1 microscope. In vitro infection assays where a host was present were labeled as "infect" regardless of whether a successful infection occurred between host and parasite. Control assays where no host was added were labeled as "control". In these experiments 20 plants per sample were used and the experiments were replicated at least twice.

**Haustorium induction assay.** Four to five days old *P. japonicum* seedlings were transferred to nutrient-free 0.8% (w/v) agar medium or 0.8% (w/v) agar medium supplemented by nutrients (½MS, ½MS no N, NH$_4$NO$_3$) for a three days pre-treatment. Subsequently, seedlings were transferred to 0.8% (w/v) agar medium containing DMBQ (Sigma-Aldrich) or DMBQ with or without nutrient treatment and grown vertically for four to five days for haustorium induction. In these experiments 20 plants per sample were used and the experiments were replicated at least twice.

**Greenhouse experiments.** Ten days old *P. japonicum* seedlings were germinated in vitro as described above. The seedlings were then transferred to pots with 50:50 soil:sand ratio. *Arabidopsis* seeds were sprinkled around the *P. japonicum* seedling. The pots were placed at 25 °C and long-day conditions (16-h light:8-h dark and 100 $\mu$mol m$^{-2}$ s$^{-1}$) and 60% humidity for 1.5 months. During this time the plants were given deionized water or water supplemented with fertilizer (commercial fertilizer Blömstra 51-10-43 N-P-K at 2 ml/L). Eight to fifteen plants per sample were used and this experiment was replicated three times.

**Histological staining.** Dissected *P. japonicum* roots or *P. japonicum* roots infecting *A. thaliana* were fixed in ethanol-acetic acid (75%/25%) solution under vacuum infiltration for 5 min. Then stained with Safranin-O solution (0.1%) at 90 °C for 5 min. The root tissue was then cleared in chloral hydrate solution (chloral hydrate: glycerol: water 8:1:2) for two to three days before observation with a Zeiss Axioscope A1 microscope[13].

**EdU and ROS staining**. Five days old *P. japonicum* seedlings were treated with water or 10.3 mM NH$_4$NO$_3$ for three days before host addition (*Arabidopsis* Col-0). 20 plants per sample were collected at 0 and 24 hpi, these experiments were replicated twice. EdU (Click-iT™ EdU Cell Proliferation Kit, Invitrogen) staining was used for the estimation of cell division. Briefly, *P. japonicum* roots were incubated in 10 μM EdU for 30 min at 25 ºC followed by tissue fixation and permeabilization following the manufacturer's instructions. EdU detection was performed with confocal microscopy (Zeiss LSM780).

For hydrogen peroxide staining 2′,7′-dichlorodihydrofluorescein diacetate CM-H$_2$DCFDA (excitation/emission 492 nm/517 nm; ThermoFisher™, C6827) was used and for superoxide staining dihydroethidium (DHE) (excitation/emission 510 nm/595 nm; Sigma-Aldrich, D7008) staining was used, *P. japonicum* roots were incubated in 10 μM CM-H$_2$DCFDA or 30 μM DHE solution in 50 mM PBS for 30 min in the dark with gentle shaking, followed by three times washing with 50 mM PBS. CM-H$_2$DCFDA or DHE detection was performed using confocal microscopy (Zeiss LSM780). Fluorescent intensity measurements were taken using ImageJ.

**Xylem strand measurement**. Five days old *P. japonicum* seedlings ($n = 19$) were treated with 1 μM ABA or 5 μM ABA, ½MS no N, ½MS or 5 mM NH$_4$NO$_3$ for three days. Afterwards, the number of xylem strands were measured at 2 mm from the root tip with a Zeiss Axioscope A1 microscope.

**Sample preparation for RNAseq**. 40 four to five days old *P. japonicum* seedlings were transferred to nutrient-free 0.8% (w/v) agar medium or 0.8% (w/v) agar medium supplemented with 10.3 mM NH$_4$NO$_3$ or 0.08 μM BA for 3 days prior to infection with *Arabidopsis* Col-0. As a control group, 40 *P. japonicum* seedlings per treatment (water, 10.3 mM NH$_4$NO$_3$ or 0.08 μM BA) remained without the *Arabidopsis* host. For the water treatment infect and control samples, five time points were prepared (0,12, 24, 48, 72 hpi). For the NH$_4$NO$_3$ and BA treatments, infect and control samples were prepared for three time points (0, 12, 24 hpi). One to two mm from *P. japonicum* and *Arabidopsis* root tips were harvested for the control plants and the 0 hpi infect plants. For the 12, 24, 48, 72 hpi time points, the haustorium, including 1–2 mm above and below tissue was collected together with the corresponding region of the *Arabidopsis* root. Three biological replicates were prepared for this experiment. RNA extraction was performed using the ROTI®Prep RNA MINI (Roth) kit following the manufacturer's instructions. The isolation of mRNA and library preparation were performed using NEBNext® Poly(A) mRNA Magnetic Isolation Module (#E7490), NEBNext® Ultra™ RNA Library Prep Kit for Illumina® (# E7530L), NEBNext® Multiplex Oligos for Illumina® (#E7600) following the manufacturer's instructions. The libraries were then sequenced using paired end sequencing with an Illumina NovaSeq 6000.

**Bioinformatic analysis**. The adapter and low-quality sequences were removed using the fastp software with default parameters[67]. The quality-filtered reads were mapped to both the *P. japonicum*[68] and *Arabidopsis* genome (TAIR10) using STAR[69] and were separated based on mapping to *P. japonicum* and *Arabidopsis* reads. The separated reads were then re-mapped to their respective genomes. The read count was calculated using FeatureCounts[70]. The differential expression analysis was used to identify differentially expressed (DE) genes between treatments and time points and was performed using Deseq2 with the default settings and q-value < 0.05[71] (Supplementary Data 2–8). The gene expression clustering was performed using the Mfuzz software[72]. Custom annotations of the *P. japonicum* predicted proteins[68] were estimated using InterProScan[73], these were used for Gene ontology analysis that was performed using the topGO software[74]. ABA and cytokinin responsive genes in *P. japonicum* (Supplementary Data 9) were identified using the tBLASTp and tBLASTp algorithm of the *Arabidopsis* ABA and cytokinin responsive genes described by[75] or *Arabidopsis* genes responsive to SA described in[76] against the *P. japonicum* genome[68]. Cell cycle and ROS related genes (Supplementary Data 9) were identified using the tBLASTp and tBLASTn algorithm of the *Arabidopsis* cell cycle and ROS related genes described[77,78] in against the *P. japonicum* genome[68].

**Statistics**. Statistical analyses were performed using one-way ANOVA followed by Tukey's HSD post-hoc test. The results of this statistical analysis are represented by compact letter display; treatments with different letters are significantly different with p-value<0.05 whereas treatments with the same letter/letters are not significantly different. For haustoria per *P. japonicum* and xylem bridge formation percentage data, the statistical analyses were performed on the means of at least 2 biological replicates, where each biological replicate consisted of 20 plants. For single comparisons, two tailed student's t-tests was used.

**qPCR**. *P. japonicum* seedlings were grown for five days before transferring to nutrient-free 0.8% (w/v) agar medium or 0.8% (w/v) agar medium supplemented with ½MS, ½MS no N or 5 μM ABA for 5 days. Additionally, *P. japonicum* seedlings were placed on pots containing 100:0, 50:50, 33:66, 25:75 soil:sand ratios. The pots were placed at 25ºC and long-day conditions (16 h light:8 h dark and 100 μmol m$^{-2}$ s$^{-1}$) and 60% humidity for 1.5 months. During this time the plants were provided deionized water. The seedlings or the shoots and roots of the above

described *P. japonicum* were then harvested and RNA extraction was performed using the ROTI®Prep RNA MINI (Roth) kit following the manufacturer's instructions. The extracted RNA was then treated with DNase I (Thermo Scientific™) following the manufacturer's instructions. cDNA synthesis was performed using Maxima First Strand cDNA Synthesis Kit for RT-qPCR (Thermo Scientific™) following the manufacturer's instructions. *PjPTB*[14] was used as an internal control. qPCR was performed with SYBR-Green master mix (Applied Biosystems™). The relative expression was calculated using the Pfaffl method[79]. All experiments were repeated at least three times with at least two technical replications each. For statistical analysis, the student's t-test was used. The primers used for this experiment are listed in Supplementary Data 10. 4 plants per sample were collected and these experiments were replicated three times.

**Hormonal quantifications**. *P. japonicum* seedlings were grown for four to five days before transferring to nutrient-free 0.8% (w/v) agar medium or 0.8% (w/v) agar medium supplemented with 10.3 mM NH$_4$NO$_3$ for three days. *Arabidopsis* Col-0 or *aba2-1* was placed next to the *P. japonicum* seedlings and left for 10 days. *P. japonicum* seedlings without a host were used as control. After 10 days with or without the presence of a host, four entire *P. japonicum* seedlings per sample and four to five entire *Arabidopsis* seedlings per sample were collected. For the mature *P. japonicum* root measurements, ~1-month-old *P. japonicum* was transferred to nutrient-free 0.8% (w/v) agar medium or 0.8% (w/v) agar medium supplemented with 10.3 mM NH$_4$NO$_3$ for seven days before *Arabidopsis* Col-0 addition. 4 roots per sample were collected at 10 dpi. The samples were crushed to powder using liquid N with mortar and pestle. Samples were extracted, purified and analyzed according to a previously published method[80]. Approximately 20 mg of frozen material per sample was homogenized and extracted in 1 mL of ice-cold 50% aqueous acetonitrile (v/v) with the mixture of $^{13}$C- or deuterium-labeled internal standards using a bead mill (27 hz, 10 min, 4 °C; MixerMill, Retsch GmbH, Haan, Germany) and sonicator (3 min, 4 °C; Ultrasonic bath P 310 H, Elma, Germany). After centrifugation (20000 × g, 15 min, 4 °C), the supernatant was purified as following. A solid-phase extraction column Oasis HLB (30 mg 1 cc, Waters Inc., Milford, MA, USA) was conditioned with 1 ml of 100% methanol and 1 ml of deionized water (Milli-Q, Merck Millipore, Burlington, MA, USA). After the conditioning steps each sample was loaded on SPE column and flow-through fraction was collected together with the elution fraction 1 ml 30% aqueous acetonitrile (v/v). Samples were evaporated to dryness using speed vac (SpeedVac SPD111V, Thermo Scientific, Waltham, MA, USA). Prior LC-MS analysis, samples were dissolved in 40 μL of 30% acetonitrile (v/v) and transferred to insert-equipped vials. Mass spectrometry analysis of targeted compounds was performed by an UHPLC-ESI-MS/MS system comprising of a 1290 Infinity Binary LC System coupled to a 6490 Triple Quad LC/MS System with Jet Stream and Dual Ion Funnel technologies (Agilent Technologies, Santa Clara, CA, USA). The quantification was carried out in Agilent MassHunter Workstation Software Quantitative (Agilent Technologies, Santa Clara, CA, USA). These experiments were repeated 4 times.

**Plasmid construction and *P. japonicum* transformation**. Plasmid construction was done using modules of Greengate cloning[81] (Addgene). For the entry modules, the *pMAS* promoter, *terMAS* terminator, and *DsRed* reporter cassette were amplified from pAGM4723 using primers with the addition of *BsaI* restriction sites and respective overhangs on the 5′ends (Supplementary Data 10), then inserted into Greengate modules pGGA000, pGGE000, and pGGF000, to create pGGA-pMAS, pGGE-terMAS, and pGGF-DsRed modules, respectively. The CDS of *Atabi1-1* was amplified from the cDNA of *Arabidopsis abi1-1* mutant, then inserted into pGGC000 to create pGGC-abi1. All of the restriction and ligation reactions were done using BsaI-HF and T4 ligase (NEB), respectively. The resulted plasmids were transformed into *E. coli* using chemically competent cells (Subcloning Efficiency™ DH5α Competent Cells, ThermoFisher Scientific) according to the manufacturer's protocol. The transformed cells were cultured and selected on LB medium with 100 μg/mL ampicillin. The plasmids were extracted using Plasmid DNA Miniprep Kit (ThermoFisher), and the sequences were confirmed by sequencing of the ligation sites (Macrogen).

To create the final binary vector pGG-*abi1*, the Greengate reaction was performed using the previously described protocol[81] using the entry vectors pGGA-pMAS, pGGB003, pGGC-abi1, pGGD002, pGGE-terMAS, pGGF-DsRed, and the empty destination vector pGGZ001. The reaction product was used for *E. coli* transformation, then the cells were cultured on LB medium with 100 μg/mL spectinomycin. The sequence was initially confirmed by digestion analysis, then sequencing of the ligation site. The plasmid was then inserted in electrocompetent *Agrobacterium rhizogenes* strain AR1193 then the cells were cultured on LB medium with 100 μg/mL spectinomycin and 50 μg/mL rifampicin.

*P. japonicum* transformation was performed according to a previously published method[36]. Briefly, three to four-day-old *P. japonicum* seedlings were sonicated for 10 to 15 seconds followed by vacuum infiltration for 5 minutes with suspension of *Agrobacterium rhizogenes* strain AR1193 carrying the overexpressing *pMAS::Atabi1-1* construct. The seedlings were then transferred on co-cultivation media (Gamborg's B5 medium, 0.8% agar, 1% sucrose, 450 μM acetosyringone) at 22 °C for 2 days in the dark conditions. Later, plants were transferred on Gamborg's B5 medium supplemented with antibiotic (0.8% agar, 1% sucrose, 300 μg/ml cefotaxime) and incubated at 25 °C under long-day conditions for

~1 month. Hairy roots expressing the construct were identified by red fluorescence using a Leica M205 FA fluorescence stereo microscope. Roots were selected based on their florescent status before being placed on nutrient-free 0.8% (w/v) agar medium or 0.8% (w/v) agar medium supplemented by ½MS or ½MS no N for 3 days prior to *Arabidopsis* (Col-0) host application. Haustoria numbers were estimated at 7 dpi. This experiment was replicated three times. Construct expression levels were estimated by qPCR using *Atabi1-1* primers (Supplementary Data 10).

**Striga-rice Infection in the rhizotron system**. The rice infection was performed in a rhizotron system[63]. 7-day-old rice seedlings were transferred to the rhizotron (10-cm × 14-cm-square petri dish with top and bottom perforation for shoot growth and water draining, filled with same size of rockwool [Nichiasu, Tokyo, Japan] onto which a 100 μm nylon mesh was placed) and fertilized with 25 mL half-strength Murashige & Skoog media per rhizotron. The root parts of the rhizotron were covered with aluminum foil and placed vertically in a growth chamber at 12-h light: 28 ºC /12-h dark: 20 ºC cycles for 2 weeks before *S. hermonthica* infection. Rice seedlings were inoculated with *S. hermonthica* seeds by placing Strigol-treated *S. hermonthica* carefully along rice roots with 5 mm intervals. Each rhizotron was inoculated by 20-60 *S. hermonthica* seeds. The rhizotron containing inoculated rice seedlings were incubated in the growth chamber described above, and developmental stages of *S. hermonthica* were categorized with a stereomicroscope (Zeiss Stemi 2000-C) after 2 and 4 weeks. Successful infection rates were calculated by the number of *S. hermonthica* with more than three leaf pairs divided by the total infected *S. hermonthica* seeds. Each rhizotron was watered with 25 mL of indicated nutrient or chemical containing solutions two times per week. The chemical concentrations used in this study were as following; 10.3 mM ammonium nitrate, 1.09 mM monosodium phosphate, 20.6 mM potassium nitrate, 0.62 mM monopotassium phosphate, 10 μM gibberellic acid, 0.08 nM 6-benzylaminopurine, 10 μM paclobutrazol, 10 or 100 μM fluridone, and 10 or 100 μM abscisic acid, 10.3 mM NH$_4$Cl, 19.69 mM KNO$_3$, 500 nM IAA or 500 nM NAA. 8 plants per sample were used and these experiments we replicated at least 3 times.

**Reporting summary**. Further information on research design is available in the Nature Research Reporting Summary linked to this article.

## Data availability

Sequence data are available at the Gene Expression Omnibus under accession numbers GSE177484. Sequence data of the *Phtheirospermum japonicum* genes studied in this article are available in GenBank (http://www.ncbi.nlm.nih.gov/genbank/) under the accession numbers provided in Supplementary Data 11. Other experimental data shown in this study are provided in the Source Data file.

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

## Acknowledgements

The authors thank Annelie Carlsbecker and Prashanth Ramachandran for providing *abi1-1C* seeds, and Thomas Spallek for critical reading of the manuscript. A.K., M.L., T.S. and C.W.M. were supported by a Wallenberg Academy Fellowship (2016-0274) and an ERC starting grant (GRASP- 805094). K.L. and J.S. were supported by grants from the Swedish Research Council, the Swedish Governmental Agency for Innovation Systems and the Knut and Alice Wallenberg Foundation. S.Y., S.C., and X.Z. were supported by MEXT KAKENHI (JP20H05909), and S.C. by KAKENHI no. 19K16169. The authors acknowledge support from the Uppsala Multidisciplinary Center for Advanced Computational Science for assistance with access to the UPPMAX computational infrastructure. We also thank the Swedish Metabolomics Centre for access to instrumentation.

## Author contributions

A.K. and C.W.M. conceived the experiments. A.K., M.L., X.Z., T.S., and J.S. performed the experiments. S.C., S.Y., K.L., and J.S. supervised the experiments. A.K. and C.W.M. wrote the paper. All authors edited and revised the final paper.

## Funding

## Competing interests

The authors declare no competing interests.
