## [Peer Review File · Nature Communications]

Nitrogen represses haustoria formation through abscisic acid in the parasitic plant *Phtheirospermum japonicum*REVIEWER COMMENTS

Reviewer #1 (Remarks to the Author):

Haustorium formation is one of the most fascinating phenomena in parasite plants. In this manuscript, Kokla et al. found that nitrate is an exogenous factor affecting *Phtheirospermum japonicum* (Pj) haustorial formation. In addition, they demonstrated that nitrate influenced the ABA content in Pj, and by exogenously applying ABA or ABA biosynthesis inhibitor, it was found that ABA had an inhibitory effect on haustorial formation. Thus, the authors conclude that nitrate is a key cue for haustorial formation of Pj. This study reveals an interesting new environmental factor that controls the formation of haustoria.

Major comments

The major conclusions should be experimentally examined using genetically modified Pj, given that transformation of Pj is available. For example, by silencing or knocking-out ABA biosynthetic genes or signaling genes, the authors could examine the function of ABA pathway in regulating haustorial formation. Furthermore, even though interesting phenotypes were found, the authors provided little information on the mechanistic level.

Other comments

1. To be specific, replace *Phtheirospermum* with *P. japonicum*.
2. L69-70: New evidence has indicated that proteins are also exchanged between parasites and hosts (Liu et al., *Mol Plant*, 2020, 13, 573-585).
3. L126: "*Phtheirospermum*-*Arabidopsis* soil infections". I did not see any *Arabidopsis* in Fig. S1C.
4. The experiment design in Fig. 2a is very complex and hard to understand. What does "Not infecting" mean? Does "We compared infected NH₄NO₃ samples with not infected NH₄NO₃ samples..." (L181-182) mean that two groups of Pj were both treated with N but one group was not given *Arabidopsis* host, whereas the group was given? But how did the authors stop Pj from infecting the hosts? It is very difficult to interpret the data without much more detailed description of the experiment design.
5. Lines 180-188. If I understood it correctly, here the Pj plants were all given N, but in one group they had infected *Arabidopsis*, but not in the other group. It is very surprising that these two groups of plants showed only around 70 DEGs at any times, considering that successful parasitization on hosts or not should have a huge impact on Pj's physiology. The authors proposed that "This low number suggested that nitrates blocked the vast majority of haustorial-related gene activation and stopped haustorial formation early during infection". If under N treatment, haustorium-related genes were not activated and stopped haustorial formation, how did the authors get the "infected NH₄NO₃ samples"? It is very confusing here.
6. L186-188, even though both groups of Pj parasites were infected on hosts, but NH₄NO₃ treatment had a very strong impact on the transcriptome of Pj, and it was shown that three genes were not activated by the N treatment. The authors found this to be "consistent with nitrate acting early to block haustoria induction". I could not follow the logic here.
7. Figure 2C, more than 10000 DEGs were detected at 72 h. This is a very large number. How were DEGs identified? Please describe it in the MM.
8. In Figure 4C & 4D, please provide the ABA data from the samples of "Pj+aba2-1" after nitrate treatment.
9. The supplemental tables do not have gene annotation information; the authors could also highlight the important genes, including those mentioned in the main text (e.g., PjYUC3, PjWOX4 and PjPMEI9), or provide another table which contains the information of these important genes, such as gene IDs.

Reviewer #2 (Remarks to the Author):

Original results are presented in this manuscript about the influence of nutrient availability on the formation of the specific organ of parasitic plants, the haustorium. They show the negative effect of N at mM levels on haustoriogenesis in the hemiparasites *Phteirospermum japonicum* and *Striga hermonthica*, notably on xylem bridge formation, and demonstrate that the N-related inhibition results from a down-regulation of genes associated to early haustorium development and an increased ABA response. This is in concordance with the known inhibiting effect of fertilizers on infestations in the fields.

For most of the experiments, the authors use *Phteirospermum japonicum* as a study topic because of their knowledge of this model (in vitro cultures of the pathosystem, haustorium induction using DMBQ, omics analyses). As a facultative parasite, comparisons between parasitic roots developing haustoria versus non parasitic ("typical, without haustoria) roots (parasitic plants versus non parasitic plants) are also of interest. In addition, since *P. japonicum* parasitizes *Arabidopsis thaliana*, the authors use appropriately some *Arabidopsis* mutants to deepen questioning.

The questions raised and the results discussed testify to a thorough and appropriate bibliographic analysis. In addition, the experiments carried out are clearly explained in the material and methods section. Among numerous results, the major are: Nitrogen at mM levels (present in MS medium or added in the medium as KNO_3 or NH_4NO_3) reduces parasitism of *P. japonicum* due to its negative impact on haustorium formation, and notably on the xylem bridge formation. Hormonal profiling shows that N treatment induces high ABA and SA levels in both non parasitizing and parasitizing *P. japonicum*, and prevents cytokinin accumulation in *P. japonicum*. Comparative time course RNAseq experiments (water, NH_4NO_3 , cytokinin or ABA treatments) in addition to qPCR confirmatory analyses using marker genes display the differentially expressed genes during early steps of parasitism, and the negative effect of N treatments on the expression of most of the genes associated with early steps of haustoriogenesis. They include xylem-related, DMBQ perception-related and ABA-responsive genes, but not cytokinin-responsive genes. Finally, rhizotrons and in vitro experiments carried out in parallel on *Striga hermonthica* demonstrate that this obligate hemiparasite shares also the control of infection, haustorium induction and development by N.

While these works seem of significance to the field, some important weaknesses balance strongly the acceptance of the manuscript for publication:

- NH_4NO_3 or KNO_3 are used in the experiments, but no experiments were carried out to test specifically NH_4 . For NH_4NO_3 treatments, it is thus more appropriate to discuss about the effect of nitrogen (N) rather than that of nitrate specifically. Numerous revisions are required in the manuscript.

Furthermore, in Fig. S1, KNO_3 20.6 mM, unlike NH_4NO_3 10.3 or 20.6 mM, has no effect on xylem bridge number and xylem plate area per haustorium. These discordant results are not discussed (p5, lines 144-145).

- Statistics and their interpretations are questionable, then making some false statement
Fig 1.D: group a missing; Fig 1.G: group b missing (and bars not visible for the treatments 0 and 50 μM)
Fig 2.C, 2.E, 3B, S3: statistics, bars missing? Experimental points should be clearly visible (Fig. 2E, 3B, S3G, H, S5G, H). Fig 5A: group a missing; Fig 5.B.: questionable statistics groups?; Fig. 6C: group c missing; Fig. S1, S5B, C, S6: statistics missing; Fig. S7: group b missing.

- Some results are inconsistent, not repeatable from one experiment to another. For example, Fluridone has no effect on the haustoria number in Fig. 5A, but reduces significantly the haustorium number in the Fig. 5E. In addition, results for the same control (water) are different in the Fig. 5A and 5E. This strongly questions robustness and methodology of the statistical analyses. Different experimental series resulting in discordant results?

- The results from the Fig S5B are not correctly discussed (p8, lines 233-241). Statistics are missing and results differ according to the gene and the organ. This part needs to be revised.

- Information is missing concerning the Fig. 5G. What is the meaning of 2, 3, 4, 5? This part needs clarification (figure and corresponding text).

Reviewer #3 (Remarks to the Author):

This manuscript describes that haustorium formation in facultative parasite *Phtheirospermum japonicum* is regulated by nutrient levels in soil through the changes in the amount of phytohormone abscisic acid (ABA). Precisely designed experiments have shown that nitrate among soil nutrients is an important factor for haustorium formation in parasite plants. In addition, nitrate increases the amount of endogenous ABA, which suppresses the haustorium formation in *Phtheirospermum japonicum*. This finding further supports why facultative parasitic plants are more likely parasite to the host when soil nutrients are poor. However, it has not been uncovered how the nitrate signal regulates the ABA metabolic genes through what regulatory proteins. If the molecular mechanism is clarified, this study including high-quality data will attract many readers. On the other hand, I think that current manuscript contains important insights into plant science and will be highly regarded in specialized journals.

Response to reviewer comments

- Comments and questions from reviewers
 - Response by authors
-

We thank the three reviewers for taking time to review our manuscript and appreciate their comments which have improved the manuscript. Our specific points are below.

Reviewer #1 (Remarks to the Author):

Haustorium formation is one of the most fascinating phenomena in parasite plants. In this manuscript, Kokla et al. found that nitrate is an exogenous factor affecting *Phtheirospermum japonicum* (Pj) haustorial formation. In addition, they demonstrated that nitrate influenced the ABA content in Pj, and by exogenously applying ABA or ABA biosynthesis inhibitor, it was found that ABA had an inhibitory effect on haustorial formation. Thus, the authors conclude that nitrate is a key cue for haustorial formation of Pj. This study reveals an interesting new environmental factor that controls the formation of haustoria.

Major comments

The major conclusions should be experimentally examined using genetically modified Pj, given that transformation of Pj is available. For example, by silencing or knocking-out ABA biosynthetic genes or signalling genes, the authors could examine the function of ABA pathway in regulating haustorial formation. Furthermore, even though interesting phenotypes were found, the authors provided little information on the mechanistic level.

We thank the reviewer for these comments and agree. Silencing or knocking out putative ABA homologs in *P. japonicum* is technically challenging and unclear if it will succeed given the large genome size, redundancy and limitations of transient hairy root assays. Instead, we adopted a similar approach to that of Spallek et al 2018 (PNAS) by expressing an *Arabidopsis* gene in *P. japonicum*, and used the well characterized *abi1-1* dominant mutant that blocks ABA signaling when expressed in multiple plant species. Overexpressing *Atabi1-1* in *P. japonicum* hairy roots blocked the suppressive effects of nitrogen upon haustoria formation, supporting our chemical data and central hypothesis. We feel these data provide strong evidence that ABA mediates nitrogen inhibition. With these new data in mind, we have modified the title to make it more accessible and to indicate that nitrogen can act via ABA.

To further investigate the mechanism for nitrate inhibition, we present new data that ROS accumulation and cell division are blocked during early haustoria formation using a combination of staining and transcriptomics. In the updated discussion, we propose that nitrogen blocks haustoria initiation via inhibiting ROS and cell cycle-related genes that are known to be critical for haustoria formation (Wada et al 2019; Wakatake et al 2018). We also present new data regarding the mechanistic basis for haustoria suppression by nitrate in *Striga*. We found that auxin treatment was sufficient to rescue this phenotype, demonstrating a role for auxin downstream of nitrate. Together, these additional data add insight into how nitrogen acts and suggests a broad role for nitrogen modulation hormone responses to affect early stages of haustoria initiation.

Other comments

1. To be specific, replace *Phtheirospermum* with *P. japonicum*.

Phtheirospermum has been replaced throughout with *P. japonicum*.

2. L69-70: New evidence has indicated that proteins are also exchanged between parasites and hosts (Liu et al., Mol Plant, 2020, 13, 573-585).

This information and citation were added at L.61.

3. L126: “*Phtheirospermum*-*Arabidopsis* soil infections”. I did not see any *Arabidopsis* in Fig. S1C.

The labels of Fig.S1c have been moved and the *Arabidopsis* plants on the image are now denoted by an arrow.

4. The experiment design in Fig. 2a is very complex and hard to understand. What does “Not infecting” mean? Does “We compared infected NH₄NO₃ samples with not infected NH₄NO₃ samples...” (L181-182) mean that two groups of *Pj* were both treated with N but one group was not given *Arabidopsis* host, whereas the group was given? But how did the authors stop *Pj* from infecting the hosts? It is very difficult to interpret the data without much more detailed description of the experiment design.

We thank the reviewer for pointing out this issue and apologize the experiment was not clearer. “Not infecting” referred to plants that had no host added; however, we have renamed this group to “control” throughout the manuscript to avoid confusion. The “infecting” samples were renamed to “infect” samples throughout the manuscript to indicate plants were given a host. Additionally, we updated the “Sample preparation for RNAseq” materials and methods section to better clarify the preparation of the samples for this experiment. The cartoon in Fig. 2a was mistakenly drawn to include the host and this was also corrected. We believe these changes will help the reader better understand the experimental design and comparisons.

5. Lines 180-188. If I understood it correctly, here the *Pj* plants were all given N, but in one group they had infected *Arabidopsis*, but not in the other group. It is very surprising that these two groups of plants showed only around 70 DEGs at any times, considering that successful parasitization on hosts or not should have a huge impact on *Pj*'s physiology. The authors proposed that “This low number suggested that nitrates blocked the vast majority of haustorial-related gene activation and stopped haustorial formation early during infection”. If under N treatment, haustorium-related genes were not activated and stopped haustorial formation, how did the authors get the “infected NH₄NO₃ samples”? It is very confusing here.

We believe that our previous issue with explaining the experiment design contributed to this query. We have now clarified our text to indicate “control” and “infect” samples. Both were given nitrogen and only 70 DEGs were different between these groups since the “infect” group had essentially no haustoria formed. We now use the term “infect” to refer to an infection having been set up (plants placed near one another) but not necessarily the formation of haustoria. We have better clarified this aspect in our materials and methods to help avoid confusion.

6. L186-188, even though both groups of *Pj* parasites were infected on hosts, but NH₄NO₃ treatment had a very strong impact on the transcriptome of *Pj*, and it was shown that three

genes were not activated by the N treatment. The authors found this to be “consistent with nitrate acting early to block haustoria induction”. I could not follow the logic here.

To help clarify the logic in L186-187 (new L.187-189), we added a Fig.1 reference in L.186. In Fig.1 we showed that nitrogen blocks haustoria formation prior to host attachment, thus early haustoria development stages. Our transcriptome data showed the transcription of genes that are upregulated during haustoria formation had decreased expression levels in nitrogen treated *P. japonicum* (Fig.3). The gene expression results shown in Fig.3 are consistent with the haustoria phenotype shown in Fig.1.

7. Figure 2C, more than 10000 DEGs were detected at 72 h. This is a very large number. How were DEGs identified? Please describe it in the MM.

The DEGs were identified from the DE analysis using Deseq2. This information has been updated and better described in the materials and methods section L.506-508.

8. In Figure 4C & 4D, please provide the ABA data from the samples of “Pj+aba2-1” after nitrate treatment.

We thank the reviewer for this useful suggestion. We repeated the hormonal quantification experiment to include “Pj+aba2-1” samples and include the quantification of additional hormones. These data are shown in updated Fig.5b,c and Fig.S5i.

9. The supplemental tables do not have gene annotation information; the authors could also highlight the important genes, including those mentioned in the main text (e.g., PjYUC3, PjWOX4 and PjPMEI9), or provide another table which contains the information of these important genes, such as gene IDs.

We thank the reviewer for pointing this out. The Tables S10-11 were updated to include gene IDs, the *Arabidopsis* homologs and GenBank information for the main genes in the text.

Reviewer #2 (Remarks to the Author):

Original results are presented in this manuscript about the influence of nutrient availability on the formation of the specific organ of parasitic plants, the haustorium. They show the negative effect of N at mM levels on haustoriogenesis in the hemiparasites *Phtheirospermum japonicum* and *Striga hermonthica*, notably on xylem bridge formation, and demonstrate that the N-related inhibition results from a down-regulation of genes associated to early haustorium development and an increased ABA response. This is in concordance with the known inhibiting effect of fertilizers on infestations in the fields. For most of the experiments, the authors use *Phtheirospermum japonicum* as a study topic because of their knowledge of this model (in vitro cultures of the pathosystem, haustorium induction using DMBQ, omics analyses). As a facultative parasite, comparisons between parasitic roots developing haustoria versus non parasitic (“typical, without haustoria) roots (parasitic plants versus non parasitic plants) are also of interest. In addition, since *P. japonicum* parasitizes *Arabidopsis thaliana*, the authors use appropriately some *Arabidopsis* mutants to deepen questioning.

The questions raised and the results discussed testify to a thorough and appropriate bibliographic analysis. In addition, the experiments carried out are clearly explained in the material and methods section. Among numerous results, the major are: Nitrogen at mM levels (present in MS medium or added in the medium as KNO₃ or NH₄NO₃) reduces parasitism of *P. japonicum* due to its negative impact on haustorium formation, and notably on the xylem bridge formation. Hormonal profiling shows that N treatment induces high ABA

and SA levels in both non parasitizing and parasitizing *P. japonicum*, and prevents cytokinin accumulation in *P. japonicum*. Comparative time course RNAseq experiments (water, NH_4NO_3 , cytokinin or ABA treatments) in addition to qPCR confirmatory analyses using marker genes display the differentially expressed genes during early steps of parasitism, and the negative effect of N treatments on the expression of most of the genes associated with early steps of haustoriogenesis. They include xylem-related, DMBQ perception-related and ABA-responsive genes, but not cytokinin-responsive genes. Finally, rhizotrons and in vitro experiments carried out in parallel on *Striga hermonthica* demonstrate that this obligate hemiparasite shares also the control of infection, haustorium induction and development by N.

While these works seem of significance to the field, some important weaknesses balance strongly the acceptance of the manuscript for publication:

1. NH_4NO_3 or KNO_3 are used in the experiments, but no experiments were carried out to test specifically NH_4 . For NH_4NO_3 treatments, it is thus more appropriate to discuss about the effect of nitrogen (N) rather than that of nitrate specifically. Numerous revisions are required in the manuscript. Furthermore, in Fig. S1, KNO_3 20.6 mM, unlike NH_4NO_3 10.3 or 20.6 mM, has no effect on xylem bridge number and xylem plate area per haustorium. These discordant results are not discussed (p5, lines 144-145).

We thank the reviewer for highlighting this important aspect. We have addressed this issue in two ways. Firstly, we performed infection assay experiments with NH_4Cl and NaNO_3 to distinguish between the effect on ammonium and nitrate. In *P. japonicum* and *Striga* both ammonium and nitrate reduced haustoria numbers with ammonium having a stronger effect (Fig.1d,e, Fig.7c,g). Secondly, we have used “nitrogen” instead of “nitrate” throughout the text and figures since both nitrate and ammonium suppress haustoria.

With 20.6mM or 10.3mM NH_4NO_3 there were no xylem bridge anatomy measurements taken since there were no or too few (<5) xylem bridges formed (Fig.1e), thus no values for these treatments in Fig.S1e,f,g. KNO_3 treatment also resulted in few xylem bridges but enough to take xylem bridge anatomy measurements. To help the reader understand this aspect, we have updated Fig.S1e,f,g with the number of measurements taken in each treatment and updated the text in L.144-146 to clarify this aspect.

2. Statistics and their interpretations are questionable, then making some false statement Fig 1.D: group a missing; Fig 1.G: group b missing (and bars not visible for the treatments 0 and 50 μM); Fig 2.C, 2.E, 3B, S3: statistics, bars missing? Experimental points should be clearly visible (Fig. 2E, 3B, S3G, H, S5G, H). Fig 5A: group a missing; Fig 5.B.: questionable statistics groups?; Fig. 6C: group c missing; Fig. S1, S5B, C, S6: statistics missing; Fig. S7: group b missing.

We have divided up this response into various aspects that we address as follows:

Fig 1.D: group a missing; The letters above the bars represent the ANOVA statistical analysis followed by Tukey's HSD test (compact letter display). If two treatments have common letters then there is no significant difference between those two treatments. On the other hand, if all letters differ between two treatments then the difference between those treatments is significant. The materials and methods L.519-522 has been updated to clarify this to the reader.

Fig 1.G: group b missing (and bars not visible for the treatments 0 and 50 μM); We have updated Fig.1g to have the error bars clearly visible. The letters above the bars represent the ANOVA statistical analysis followed by Tukey's HSD test (compact letter display). The materials and methods L.519-522 has been updated to clarify this to the reader.

Fig 2.C, 2.E, 3B, S3: statistics, bars missing? Fig.2C (new Fig.S3e) refers to absolute numbers of DE genes that do not vary and thus there are no error bars, Fig.2E (new Fig.2c) we have updated and included error bars representing standard deviation, Fig.3B (new Fig.3b) we have updated and included error bars representing standard deviation, Fig.S3 (new Fig.S3b-h) refers to absolute numbers of DE genes that do not vary and thus there are no error bars, Fig.S3 (new Fig.S3i, j) we have updated and included error bars representing standard deviation.

Experimental points should be clearly visible (Fig. 2E (new Fig.2c), 3B (new Fig.3b), S3G, H (new Fig.S3i, j), S5G, H (new Fig.S5d, e)). We have updated the x-axis time points which have been re-aligned to be clearly visible.

Fig 5A: group a missing; new Fig.6a, we have updated and simplified the ANOVA representation by compact letter display.

Fig 5.B.: questionable statistics groups? New Fig.6b, we have updated and simplified the ANOVA representation by compact letter display.

Fig. 6C: group c missing; new Fig.7b, the letters above the bars represent the ANOVA statistical analysis followed by Tukey's HSD test (compact letter display). The materials and methods L.519-522 has been updated to clarify this to the reader.

Fig. S1, S5B, C, S6: statistics missing; We updated Fig.S1, Fig.S5b,c and Fig.S6 to include error bars and statistical analyses.

Fig. S7: group b missing. The letters above the bars represent the ANOVA statistical analysis followed by Tukey's HSD test (compact letter display). The materials and methods L.519-522 has been updated to clarify this to the reader.

3. Some results are inconsistent, not repeatable from one experiment to another. For example, Fluridone has no effect on the haustoria number in Fig. 5A, but reduces significantly the haustorium number in the Fig. 5E. In addition, results for the same control (water) are different in the Fig. 5A and 5E. This strongly questions robustness and methodology of the statistical analyses. Different experimental series resulting in discordant results?

Parasitic plant infection assays can have some variability but the trends are generally robust. However, we agree with the reviewer that 5A and 5E show differences for fluridone. We repeated this assay several more times and have updated Fig.5a (new Fig.6a). In Fig.6a the difference between water and FI is not significant but it is significant in Fig.6c. This is due to the small reduction in haustoria numbers in response to fluridone treatment and the variation in the water treatment in Fig.6a. Due to this variation and slight haustoria reduction in the FI treatment the statistical test renders this difference not significant.

The water treatments in Fig.6a,c do not differ significantly (p -value=0.2) and the letters represent the results of the ANOVA statistical analysis. This analysis is performed separately for the data shown in each graph. Thus the "letters" represent comparisons between treatments within an individual graph and not across different graphs.

4. The results from the Fig S5B are not correctly discussed (p8, lines 233-241). Statistics are missing and results differ according to the gene and the organ. This part needs to be revised.

We have updated Fig.S5b and included statistical analysis. We accordingly updated the text to clarify these results.

5. Information is missing concerning the Fig. 5G. What is the meaning of 2, 3, 4, 5? This part needs clarification (figure and corresponding text).

We thank the reviewer for pointing out this omission. Fig.5g (new Fig.S6j,k) have been updated to clarify that the numbers 2, 3, 4, 5 refer to the number of xylem strands.

Reviewer #3 (Remarks to the Author):

This manuscript describes that haustorium formation in facultative parasite *Phtheirospermum japonicum* is regulated by nutrient levels in soil through the changes in the amount of phytohormone abscisic acid (ABA). Precisely designed experiments have shown that nitrate among soil nutrients is an important factor for haustorium formation in parasite plants. In addition, nitrate increases the amount of endogenous ABA, which suppresses the haustorium formation in *Phtheirospermum japonicum*. This finding further supports why facultative parasitic plants are more likely parasite to the host when soil nutrients are poor. However, it has not been uncovered how the nitrate signal regulates the ABA metabolic genes though what regulatory proteins. If the molecular mechanism is clarified, this study including high-quality data will attract many readers. On the other hand, I think that current manuscript contains important insights into plant science and will be highly regarded in specialized journals.

We thank the reviewer for the critical reading of our work. We have substantially revised the manuscript and now include genetic evidence that ABA signalling is required for the nitrogen inhibition of haustoria. We also provide evidence that nitrogen suppresses early ROS accumulation and cell division, demonstrating that nitrogen limits haustoria formation by blocking early processes previously shown to be important for haustoria formation (Wada et al 2019; Wakatake et al 2018). We also provide additional evidence regarding the mechanism for nitrogen suppression of *Striga* haustoria formation. There, auxin treatment is sufficient to restore haustoria formation indicating that auxin acts downstream of nitrogen. Given the limited genetic resources and tools in *Striga* and *P. japonicum*, we feel our manuscript provides important insight into how parasitic plants monitor their environment and regulate haustoria numbers according to nitrogen availability, and that these effects are mediated by auxin and ABA.

REVIEWER COMMENTS

Reviewer #1 (Remarks to the Author):

Comments to the authors

The authors revised manuscript substantially and improved it significantly, especially by supplementing new data, in which the Atabi1-1 was overexpressing in *P. japonicum* hairy roots to genetically examine the function of ABA signaling in haustorial formation. In addition, some of the suggested changes, concerns, corrections have been implemented and addressed accordingly. However, revision are still needed.

1. The experiment design in Figure 2 is still hard to understand. The exact design should be described much more in detail in the main text ("Haustoria formation induces widespread transcriptional changes") or in the figure caption. Even though there is some information of the experiment design in the MM, it would be helpful to let the readers know here in the main text or the figure caption. If I understood correctly, there were three treatment groups, water-agar, NH₄NO₃, and BA treatment. Did each group have a "control" and "infect" treatment again (please describe the purpose of having "control" and "infect")? Which groups exactly were used for transcriptome analysis and how were the "control" and "infect" compared? It seems that Fig. 2b only included the "infect" "water-agar" group, and if so, what was the "control" used for?
2. The Figure S3 was called out earlier than was Figure S2. Please adjust their order.
3. Lines 230-231 "P. japonicum ABA levels were reduced in aba2-1 infections compared to Col-0 infections, suggesting some host-derived ABA moved to the parasite". Here a possibility that the reduced ABA levels in Pj was due to certain communication between host and Pj's ABA signaling but not ABA movement cannot be ruled out.
4. Figure 4, what does "DHE and H2DCFDA" mean? Their full names should be given in the figure caption and in L325.

Reviewer #2 (Remarks to the Author):

The author's answers are generally satisfactory in raising my questions/remarks on the major points from the original manuscript, notably concerning statistics. I nevertheless remain questioning the choice of the authors to present results of independent experiments with certain repeated treatments (water and fluridone treatments, Fig. 6a, c) showing differences in statistics. The complementary experiments carried out in addition to the revisions of the text (notably in the results part) greatly improve the manuscript.

However, I have some remarks/comments/suggestions on this revised version. In my opinion, some of them needs a specific response from the authors (interpretations of the results) and probably revisions in the text.

:

1. Introduction

- p3 lines 126-127: redundant information from lines 109-110 .
- p4 line 154-155: I propose for this sentence: " ... reducing parasite development instead of only "parasite germination."

2. Results

- p5 line 213: "Fig 1 a-c" instead of "Fig a-b"
- p5 lines 226-229: KCl treatment is missing to eliminate the hypothesis of a possible chlorine effect for NH₄Cl treatment. The hypothesis that NH₄ has a stronger effect than NO₃ is questionable.
- p5 lines 232: The sentence should be corrected to be in agreement with the results and statistics: "a 5 to 20.6 mM range of concentrations" instead of "a wide range of concentrations ranging from 50 μM to 20.6 mM)

- p5 line 233: Fig. S1e also?
- p6 lines 332-337: the paragraph should be corrected. The results (DMBQ experiments) show that nitrogen affects the formation of the haustorium but do not, in my opinion, rule out the hypothesis that nitrogen could also stimulate the host's defences.
- p9 lines 711-716: poor soil experiments: The authors suggest that nutrient deficiency reduces the response to ABA, based on some of their findings (overexpression of two ABA response genes). This suggestion is contradicted by the over-expression of the other two ABA response genes also tested in this study. Results do not fully support this suggestion.
- p9 lines 716-717: "SA levels also increased during N treatment". Please specify in the parasite, in the host, in both?
- p9 line 716-719 and p10 lines 773-776: a conclusion/suggestion regarding SA is expected.
- p9 line 776: Fig. 6c, d, e also?
- p9 line 782: Fig. 6b, e are inappropriate for "xylem bridge numbers ad size".
- p9 line 783: please add the Fig. S6g (fluridone treatment)
- p11 line 848-849: very interesting suggestion but auxin treatment on *P. japonicum* roots is missing (and should be necessary) in this study. Similar remark for the discussion part (p13 lines 1038-1040)

3. Discussion

- p12 lines 921-922: false statement. The results of this study (Fig. 1, 5a,b) do not show that nitrogen treatment reduces CK levels and CK response. This part seems to contradict previous comments (pages 8-9 lines 604-705). Please this needs clarification.
- p12 line 1027: Fig 4c : inappropriate figure for ABA levels?
- p12 line 1041-1042: "low level of ABA appeared important for xylem bridge formation. Please specify the results that lead to this suggestion.

4. Methods

- p14 line 1107: final water washes of seeds after EtOH washes?
- p14 lines 1119-1121: "conditioned" seeds instead of "preconditioned seeds". Indeed, most of the scientific community (parasitic plants) use the term of conditioning for this preliminary stage.
- p21 lines 1382-1383. Please indicate the "Striga seed inoculum" as mg seeds per rhizotron

5. Figure legends

- page 32 line 886: Comparisons to the "water treatment" instead of "to fertilizer treatment) should be more appropriate.
- page 32 line 901: "NH₄NO₃ reduces ..." instead of "Nitrogen reduces ..." should be more appropriate for the Fig. 4 legend (in relation to the fig. 3)
- p33 line 931: "g" instead of "I" for Expression levels of AtABI1-1
- p34 line 934 : not "g" for the scale bars
- p34 line 937: "Nitrogen inhibits ..." instead of "NH₄NO₃ inhibits" should be more appropriate for this figure (various N compounds tested).

Legend of the y axis (Fig. 1): "FW (g) instead of "FW (gr) ..."

6. Supplementary figures

- Fig S1. Errors in the text. The number of measurements is not indicated for the Fig. S1a and b.
- Fig S4a. Errors concerning the BA control vs water control "down regulated" instead of "up-regulated" (fourth part of this figure) and concerning the "hpi in the columns"
- Fig S4b, d: legend of the x axis: N control instead of nitrate control
- Fig S5i title: SA quantification ... should be more accurate than "hormonal quantification"

Response to reviewer comments

- Comments and questions from reviewers
 - Response by authors
-

We thank the reviewers for taking time to re-examine our revised manuscript and appreciate their comments. In our revised version, we have shorted the abstract to meet formatting requirements and modified the title. Figure 4 was re-ordered so that the panels are presented in the order they are discussed in the text. The source data file is included. Our specific points to the reviewers' comments are below.

Reviewer #1 (Remarks to the Author):

Comments to the authors

The authors revised manuscript substantially and improved it significantly, especially by supplementing new data, in which the Atabi1-1 was overexpressing in *P. japonicum* hairy roots to genetically examine the function of ABA signalling in haustorial formation. In addition, some of the suggested changes, concerns, corrections have been implemented and addressed accordingly. However, revisions are still needed.

We thank the reviewer for the critical reading of our work and their suggestions.

1. The experiment design in Figure 2 is still hard to understand. The exact design should be described much more in detail in the main text ("Haustoria formation induces widespread transcriptional changes") or in the figure caption. Even though there is some information of the experiment design in the MM, it would be helpful to let the readers know here in the main text or the figure caption. If I understood correctly, there were three treatment groups, water-agar, NH₄NO₃, and BA treatment. Did each group have a "control" and "infect" treatment again (please describe the purpose of having "control" and "infect")? Which groups exactly were used for transcriptome analysis and how were the "control" and "infect" compared? It seems that Fig. 2b only included the "infect" "water-agar" group, and if so, what was the "control" used for?

To better clarify our RNAseq experimental design we renamed "water-agar" to "water" throughout the text. The text was also modified in "Haustoria formation induces widespread transcriptional changes" and "materials and methods" paragraphs to better clarify the RNAseq experimental design. The text was modified to clarify the RNAseq comparisons done for each results section. Additionally, Fig.2a and its legend were updated.

2. The Figure S3 was called out earlier than was Figure S2. Please adjust their order.

The order of these figures was adjusted and Fig.S3 is now Fig.S2 and Fig.S2 is now Fig.S3 to respect the order of appearance on the text.

3. Lines 230-231 "P. japonicum ABA levels were reduced in aba2-1 infections compared to Col-0 infections, suggesting some host-derived ABA moved to the parasite". Here a possibility that the reduced ABA levels in Pj was due to certain communication between host and Pj's ABA signalling but not ABA movement cannot be ruled out.

The sentence was updated to include the possibility of parasite-host ABA signalling as a possible reason for the ABA reduction in *P. japonicum*.

4. Figure 4, what does “DHE and H2DCFDA” mean? Their full names should be given in the figure caption and in L325.

The Fig.4 legend and text has been updated to include the full name of DHE and H2DCFDA.

Reviewer #2 (Remarks to the Author):

The author's answers are generally satisfactory in raising my questions/remarks on the major points from the original manuscript, notably concerning statistics. I nevertheless remain questioning the choice of the authors to present results of independent experiments with certain repeated treatments (water and fluridone treatments, Fig. 6a, c) showing differences in statistics.

We thank the reviewer for taking the time to critically read our manuscript and for the detailed points – these are appreciated. The fluridone and water treatments are the combinations of multiple independent experiments which give the same trends but different statistical results. To avoid confusion, we have combined the datasets of Fig.6a, c and Fig.6b, d which increases the replicates and gives more statistical power to the ANOVA testing. It also makes Figure 6 simpler and easier for the reader to understand. The trends remain the same. The updated figure is Fig.6a, b.

The complementary experiments carried out in addition to the revisions of the text (notably in the results part) greatly improve the manuscript.

However, I have some remarks/comments/suggestions on this revised version. In my opinion, some of them needs a specific response from the authors (interpretations of the results) and probably revisions in the text:

1. Introduction

- p3 lines 126-127: redundant information from lines 109-110.

The text has been updated to remove the redundant information.

- p4 line 154-155: I propose for this sentence: “.... reducing parasite development instead of only “parasite germination.”

The text was updated as suggested.

2. Results

-p5 line 213: “Fig 1 a-c” instead of “Fig a-b”

The figure reference in was updated to Fig.1a-c.

-p5 lines 226-229: KCl treatment is missing to eliminate the hypothesis of a possible chlorine effect for NH₄Cl treatment. The hypothesis that NH₄ has a stronger effect than NO₃ is questionable.

We performed assays of *P. japonicum* infecting *Arabidopsis* Col-0 under KCl treatment and now show these data in Fig.1d, e. In our assays KCl did not affect haustoria numbers or xylem bridge formation suggesting that the haustoria reduction under NH₄Cl treatment is due to the effect of NH₄ and not the effect of Cl. We have updated the text to include KCl. We

have also removed the phrase stating the hypothesis that ammonium has a stronger effect than nitrate.

- p5 lines 232: The sentence should be corrected to be in agreement with the results and statistics: “a 5 to 20.6 mM range of concentrations” instead of “a wide range of concentrations ranging from 50 μ M to 20.6 mM)

The text was rephrased to “a 50 μ M to 20.6 mM range of concentrations” as suggested.

- p5 line 233: Fig. S1e also?

The figure reference was updated to Fig.S1e, f, g.

- p6 lines 332-337: the paragraph should be corrected. The results (DMBQ experiments) show that nitrogen affects the formation of the haustorium but do not, in my opinion, rule out the hypothesis that nitrogen could also stimulate the host's defences.

The text was rephrased to avoid this reference. We have also modified the discussion to include the possibility that nitrogen might additionally stimulate host defences or reduce host HIF production. Given our ability to modify the effects of nitrogen by blocking parasite ABA-signalling, our data are consistent with nitrogen having its main effect on the parasite.

- p9 lines 711-716: poor soil experiments: The authors suggest that nutrient deficiency reduces the response to ABA, based on some of their findings (overexpression of two ABA response genes). This suggestion is contradicted by the over-expression of the other two ABA response genes also tested in this study. Results do not fully support this suggestion.

We agree with the reviewer that only 2 of the 4 tested genes are downregulated in nutrient poor soils. We have included expression data for an additional ABA-related gene, *PjABA2*, that shows downregulation. We have rephrased our text to better explain that nutrient poor soils likely have a partial transcriptional effect on ABA response and biosynthesis.

- p9 lines 716-717: “SA levels also increased during N treatment”. Please specify in the parasite, in the host, in both?

The text was rephrased to specify that in this sentence we refer to the SA increase in the parasite under nitrogen treatment.

- p9 line 716-719 and p10 lines 773-776: a conclusion/suggestion regarding SA is expected.

The text was updated to provide a conclusion.

- p10 line 776: Fig. 6c, d, e also?

The figure reference in was updated to include Fig.S6c, d, e, h, i.

- p10 line 782: Fig. 6b, e, are inappropriate for “xylem bridge numbers ad size”.

The text was updated to “xylem bridge formation, numbers or size” followed by the (Fig.6b, e) figure reference.

- p10 line 783: please add the Fig. S6g (fluridone treatment)

The (Fig.6b, Fig.S6i) reference was added (S6g is now S6i).

- p11 line 848-849: very interesting suggestion but auxin treatment on *P. japonicum* roots is missing (and should be necessary) in this study. Similar remark for the discussion part (p13 lines 1038-1040)

To address this, we have included the results of infection assays with *P. japonicum* and Col-0 and ABA, NAA, 1/2MS, ABA+NAA, 1/2MS+NAA treatments (Fig.S7e, f) showing that auxin application did not rescue either the ABA or 1/2MS haustoria inhibitory effect in *P. japonicum*. The text was also modified accordingly.

3. Discussion

- p12 lines 921-922: false statement. The results of this study (Fig. 1, 5a, b) do not show that nitrogen treatment reduces CK levels and CK response. This part seems to contradict previous comments (pages 8-9 lines 604-705). Please this needs clarification.

We agree and have updated Figure 5 and Figure S6 to more clearly present cytokinin levels, since our previous figure made it difficult to see differences due to the low concentrations. Rather than showing N-glucosides (tZ7G and tZ9G) which are non-active CK products of irreversible deactivation, we show the active CK compound trans-zeatin and precursor trans-zeatin riboside which show upregulation in Arabidopsis upon nitrate treatment (Figure S6), upregulation in Pj upon infection (Figure 5) and are consistent with previous literature (eg. Takei et al 2004; Spallek et al 2017). The trends largely remain the same and we have rephrased the results and discussion to clarify that nitrogen reduces tZ levels and does not induce a strong CK response in young *P. japonicum*.

- p12 line 1027: Fig 4c: inappropriate figure for ABA levels?

The figure reference was updated to (Fig. 5a) to refer to the figure showing these data.

- p12 line 1041-1042: "low level of ABA appeared important for xylem bridge formation. Please specify the results that lead to this suggestion.

This section was rephrased and the following figure reference (Fig.6b, Fig.S6l) was added to refer to the figures showing these data.

4. Methods

- p14 line 1107: final water washes of seeds after EtOH washes?

The sentence was updated to explain that the seeds were left to air dry after the ethanol washes.

- p14 lines 1119-1121: "conditioned" seeds instead of "preconditioned seeds". Indeed, most of the scientific community (parasitic plants) use the term of conditioning for this preliminary stage.

The term "preconditioned" was changed to "conditioned".

- p21 lines 1382-1383. Please indicate the "Striga seed inoculum" as mg seeds per rhizotron

The text was updated to inform that 20-60 Striga seeds were used as inoculum for each rhizotron.

5. Figure legends

- page 32 line 886: Comparisons to the "water treatment" instead of "to fertilizer treatment) should be more appropriate.

Fig.1 legend was updated to “water treatment” instead of “no fertilizer treatment”. Fig.1b, c and Fig.S1a, b x-axis was also updated to “water” instead of “no fertilizer”.

- page 32 line 901: “NH₄NO₃ reduces ...” instead of “Nitrogen reduces ...” should be more appropriate for the Fig. 4 legend (in relation to the fig. 3)

The Fig.4 legend was updated to “NH₄NO₃” instead of “nitrogen”.

- p33 line 931: “g” instead of “l” for Expression levels of AtABI1-1

Fig.6 legend was updated.

- p34 line 934: not “g” for the scale bars

Fig.6 legend was updated to indicate that the scale bars correspond to (f).

- p34 line 937: “Nitrogen inhibits ...” instead of “NH₄NO₃ inhibits” should be more appropriate for this figure (various N compounds tested).

The legend of Fig.7 was updated from “NH₄NO₃” to “Nitrogen”.

Legend of the y axis (Fig. 1): “FW (g) instead of “FW (gr) ...”

The y-axis legend of Fig.1b and Fig.S1b was updated to “FW (g)...”.

6. Supplementary figures

- Fig S1. Errors in the text. The number of measurements is not indicated for the Fig. S1a and b.

The legend was updated to mention “(mean±SD, n=4-5 per treatment, 3 replicates)”.

- Fig S4a. Errors concerning the BA control vs water control “down regulated” instead of “up-regulated” (fourth part of this figure) and concerning the “hpi in the columns”

Fig.S4a was corrected to mention the time points instead of “downregulated” .

- Fig S4b, d: legend of the x axis: N control instead of nitrate control

The x-axis legend of Fig.S4b, d was updated to “NH₄NO₃” instead of “nitrate”.

- Fig S5i title: SA quantification ... should be more accurate than “hormonal quantification”

Arabidopsis SA quantifications were moved to Figure S6a and the figure legend updated to specify which hormones were quantified.

REVIEWERS' COMMENTS

Reviewer #1 (Remarks to the Author):

In the revised manuscript, the authors made many changes including modifying the title. In particular, the current description of the experimental design of RNA-seq in Figure 2 has become easier for readers to understand.

I just have some minor concerns

1. Line 273. "Fig. 6a, c" should be replaced with "Fig. 6a, b".
2. In Fig. 1d-g, the data should be multiple comparisons among different treatments by ANOVA analysis. "comparisons to the water treatment" seems to be wrong. Please rewrite the description of statistical analyses, and it is better separate b and c from d-h, as b and c use t-test but d-h use ANOVA.

Reviewer #2 (Remarks to the Author):

The authors' answers to my numerous questions or remarks, the corrections made for statistical treatments (Figure 6) and the revisions made to the text suit me. I consider them sufficient and I therefore have no comments on this last version of the manuscript.

Response to reviewer comments

- Comments and questions from reviewers
 - Response by authors
-

We thank the reviewers for taking time to re-examine our revised manuscript and appreciate their comments. Our specific points to the reviewers' comments are below.

Reviewer #1 (Remarks to the Author):

In the revised manuscript, the authors made many changes including modifying the title. In particular, the current description of the experimental design of RNA-seq in Figure 2 has become easier for readers to understand.

I just have some minor concerns

1. Line 273. "Fig. 6a, c" should be replaced with "Fig. 6a, b".

The figure reference in L. 273 was updated to "Fig.6a, b"

2. In Fig. 1d-g, the data should be multiple comparisons among different treatments by ANOVA analysis. "Comparisons to the water treatment" seems to be wrong. Please rewrite the description of statistical analyses, and it is better separate b and c from d-h, as b and c use t-test but d-h use ANOVA.

The figure legend of Figure 1 was updated to better describe that in Fig.1b, c Student's t-test was performed while in Fig.1d-h ANOVA was performed.

Reviewer #2 (Remarks to the Author):

The authors' answers to my numerous questions or remarks, the corrections made for statistical treatments (Figure 6) and the revisions made to the text suit me. I consider them sufficient and I therefore have no comments on this last version of the manuscript.

We thank the reviewer for taking the time to revise our manuscript.